# Explicit Explore-Exploit Algorithms in Continuous State Spaces

**Mikael Henaff**
Microsoft Research
`mihenaff@microsoft.com`

## Abstract

We present a new model-based algorithm for reinforcement learning (RL) which consists of explicit exploration and exploitation phases, and is applicable in large or infinite state spaces. The algorithm maintains a set of dynamics models consistent with current experience and explores by finding policies which induce high disagreement between their state predictions. It then exploits using the refined set of models or experience gathered during exploration. We show that under realizability and optimal planning assumptions, our algorithm provably finds a near-optimal policy with a number of samples that is polynomial in a structural complexity measure which we show to be low in several natural settings. We then give a practical approximation using neural networks and demonstrate its performance and sample efficiency in practice.

## 1  Introduction

What is a good algorithm for systematically exploring an environment for the purpose of reinforcement learning? A good answer could make the application of deep RL to complex problems [31, 30, 28, 20] much more sample efficient. In tabular Markov Decision Processes (MDPs) with a small number of discrete states, model-based algorithms which perform exploration in a provably sample-efficient manner have existed for over a decade [24, 5, 47]. The first of these, known as the Explicit Explore-Exploit ($E^3$) algorithm [24], progressively builds a model of the environment's dynamics. At each step, the agent uses this model to plan, either to *explore* and reach an unknown state, or to *exploit* and maximize its reward within the states it knows well. By actively seeking out unknown states, the algorithm provably learns a near-optimal policy using a number of samples which is at most polynomial in the number of states. Many problems of interest, however, have a set of states which is infinite or extremely large (for example, all images represented with finite precision), and in these settings, tabular algorithms are no longer applicable.

In this work, we propose a new $E^3$-style algorithm which operates in large or continuous state spaces. The algorithm maintains a set of dynamics models which are consistent with the agent's current experience, and explores the environment by executing policies designed to induce high disagreement between their predictions. We show that under realizability and optimal planning assumptions, our algorithm provably finds a near-optimal policy using a number of samples from the environment which is independent of the number of states, and is instead polynomial in the rank of the *model misfit matrix*, a structural complexity measure which we show to be low in natural settings such as small tabular MDPs, large MDPs with factored transition dynamics [23] and (potentially infinite) low rank MDPs. We then present a practical version of the algorithm using neural networks, and demonstrate its performance and sample efficiency empirically on several problems with large or continuous state spaces.

**Algorithm 1** $(\mathcal{M}, \Pi, n, \epsilon, \phi)$

---

1: **Inputs** Initial model set $\mathcal{M}$, policy class $\Pi$, number of trajectories $n$, tolerance $\epsilon$, model error $\phi$.
2: $\mathcal{M}_1 \leftarrow \mathcal{M}$
3: Initialize replay buffer $\mathcal{R} \leftarrow \emptyset$.
4: **for** $t = 1, 2, ...$ **do**
5:      $\pi_{\text{explore}}^t = \text{argmax}_{\pi \in \Pi} \left[ v_{\text{explore}}(\pi, \mathcal{M}_t) \right]$
6:      **if** $v_{\text{explore}}(\pi_{\text{explore}}^t, \mathcal{M}_t) > \frac{\epsilon}{|\mathcal{A}|}$ **then**
7:          Collect dataset of $n$ trajectories following $\pi_{\text{explore}}^t$, add to replay buffer $\mathcal{R}$
8:          $\mathcal{M}_{t+1} \leftarrow \texttt{UpdateModelSet}(\mathcal{M}_t, \mathcal{R}, \phi)$
9:      **else**
10:         Choose any $\tilde{M} \in \mathcal{M}_t$
11:         $\pi_{\text{exploit}} = \text{argmax}_{\pi \in \Pi} \left[ v_{\text{exploit}}(\pi, \tilde{M}) \right]$
12:         Halt and return $\pi_{\text{exploit}}$
13:      **end if**
14: **end for**

---

## 2 Algorithm

We consider an episodic, finite-horizon MDP setting defined by a tuple $(\mathcal{S}, \mathcal{A}, M^\star, R^\star, H)$. Here $\mathcal{S}$ is a set of states (which could be large or infinite), $\mathcal{A}$ is a discrete set of actions, $M^\star$ is the true (unknown) transition model mapping state-action pairs to distributions over next states, $R^\star$ is the true function mapping states to rewards in $[0, 1]$, and $H$ is the horizon length. For simplicity we assume rewards are part of the state and the agent has access to $R^\star$, so the task of predicting future rewards is included in that of predicting future states. A state $s \in \mathcal{S}$ at time step $h$ will be denoted by $s_h$.

The general form of our algorithm is given by Algorithm 1. At each epoch $t$, the algorithm maintains a set of dynamics models $\mathcal{M}_t$ which are consistent with the experience accumulated so far, and searches for an exploration policy which will induce high disagreement between their predictions. If such a policy is found, it is executed and the set of models is updated to reflect the new experience. Otherwise, the algorithm switches to its exploit phase and searches for a policy which will maximize its predicted future rewards.

Let $P_M^{\pi,h}(\cdot)$ denote the distribution over states at time step $h$ induced by sampling actions from policy $\pi$ and transitions from model $M$, and let $\mathcal{D}(\pi, M, M', h) = \delta(P_M^{\pi,h}(\cdot), P_{M'}^{\pi,h}(\cdot))$, where $\delta$ denotes a distance measure between probability distributions such as KL divergence or total variation. The quantity which the exploration policy seeks to maximize at epoch $t$ is given by:

$$v_{\text{explore}}(\pi, \mathcal{M}_t) = \max_{M, M' \in \mathcal{M}_t} \sum_{h=1}^{H} \mathcal{D}(\pi, M, M', h)$$

Maximizing this quantity can be viewed as solving a fictitious *exploration MDP*, whose state space is the concatenation of $|\mathcal{M}_t|$ state vectors in the original MDP, whose transition matrix consists of a block-diagonal matrix whose blocks are the transition matrices of the models in $\mathcal{M}_t$, and whose reward function is the distance measured using $\delta$ between the pairs of components of the state vector corresponding to different models. Importantly, searching for an exploration policy can be done internally by the agent and does not require any environment interaction, which will be key to the algorithm's sample efficiency.

Once the agent can no longer find a policy which induces sufficient disagreement between its candidate models in $\mathcal{M}_t$, it chooses a model and computes an exploitation policy using the model's predicted reward:

$$v_{\text{exploit}}(\pi, M) = \sum_{h=1}^{H} \sum_{s_h} P_M^{\pi,h}(s_h) R^\star(s_h)$$

# 3 Sample Complexity Analysis

## 3.1 Algorithm Instantiation

We first give an instantiation of Algorithm 1, called DREEM (DisagReement-led Elimination of Environment Models), for which we will prove sample complexity results. All proofs can be found in Appendix A. The algorithm starts with a large set of candidate models $\mathcal{M}$, which is assumed to contain the true model, and iteratively eliminates models which are not consistent with the experience gathered through the exploration policy. We will show that the number of samples needed to find a near-optimal policy is independent of the number of states, and is instead polynomial in $|\mathcal{A}|, H, \log|\mathcal{M}|, \log|\Pi|$, and the rank of the *model misfit matrix*, a quantity which we define below and which is low in natural settings. DREEM is identical to Algorithm 1, with the `UpdateModelSet` subroutine instantiated as follows:

---

**Algorithm 2** `UpdateModelSet`$(\mathcal{M}_t, \mathcal{R}, \phi)$

---

1: For each $M \in \mathcal{M}_t, h \leq H$, compute $\widetilde{\mathcal{W}}(\pi_{\text{explore}}^t, M, h)$ using data from $\mathcal{R}$ collected using the last exploration policy $\pi_{\text{explore}}^t$
2: $\mathcal{M}_{t+1} \leftarrow \{M \in \mathcal{M}_t : \widetilde{\mathcal{W}}(\pi_{\text{explore}}^t, M, h) \leq \phi \text{ for all } h \leq H\}$
3: Return $\mathcal{M}_{t+1}$

---

The quantity $\mathcal{W}(\pi, M, h)$ can be thought of as the error of model $M$ in parts of the state space visited by $\pi$ at time step $h$, and is formally defined below.

**Definition 1.** *The misfit of model $M$ discovered by policy $\pi$ at time step $h$ is given by:*

$$\mathcal{W}(\pi, M, h) = \mathbb{E}_{s_{h-1} \sim P_{M^\star}^{\pi, h-1}, a_{h-1} \sim U(\mathcal{A})} \left[ \|P_M(\cdot|s_{h-1}, a_{h-1}) - P_{M^\star}(\cdot|s_{h-1}, a_{h-1})\|_{TV} \right]$$

*The empirical misfit estimated using a dataset collected by following $\pi$ is denoted $\widetilde{\mathcal{W}}(\pi, M, h)$.*

See Appendix A.1 for details on computing $\widetilde{\mathcal{W}}$. We will make use of the following two assumptions:

**Assumption 1.** $\mathcal{M}$ contains the true model $M^\star$ and $\Pi$ contains optimal policies for all models in $\mathcal{M}$.

**Assumption 2.** The policy optimizations in Algorithm 1 are performed exactly.

The first is a standard realizability assumption. The second assumes access to an optimal planner and has been used in several previous works [24, 23, 22, 5]. This does not mean that the planning problem is trivial, but is meant to separate the difficulty of planning from that of exploration.

We note that DREEM will not be computationally feasible for many problems since the sets $\mathcal{M}_t$ will often be large, and the algorithm requires iterating over them during both the elimination and planning steps. However, it distills the key ideas of Algorithm 1 and demonstrates its sample efficiency when optimizations can be performed exactly. We will later give a practical instantiation of Algorithm 1 and demonstrate its sample efficiency empirically.

## 3.2 Structural Complexity Measure

Since we are considering settings where $\mathcal{S}$ is large or infinite, it is not meaningful to give sample complexity results in terms of the number of states, as is often done for tabular algorithms. We instead use a structural complexity measure which is independent of the number of states, and depends on the maximum rank over a set of error matrices, which we define next. [1]

**Definition 2.** *(Model Misfit Matrices) Let $\mathcal{M}$ be a model class and $\Pi$ a policy class. Define the set of matrices $A_1, ..., A_H \in \mathbb{R}^{|\Pi| \times |\mathcal{M}|}$ by $A_h(\pi, M) = \mathcal{W}(\pi, M, h)$ for all $\pi \in \Pi$ and $M \in \mathcal{M}$.*

Using the ranks of error matrices as complexity measures of RL environments was proposed in [21, 49]. Although the model misfit matrices $A_h$ may themselves be very large, we show next that their ranks are in fact small in several natural settings.

**Proposition 1.** *Assume $|\mathcal{S}|$ is finite and let $A_h$ be the matrix defined above. Then $rank(A_h) \leq |\mathcal{S}|$.*

**Proposition 2.** *Let $\Gamma$ denote the true transition matrix of size $|\mathcal{S}| \times |\mathcal{S} \times \mathcal{A}|$, with $\Gamma(s', (s, a)) = P_{M^\star}(s'|s, a)$. Assume that there exist two matrices $\Gamma_1, \Gamma_2$ of sizes $|\mathcal{S}| \times K$ and $K \times |\mathcal{S} \times \mathcal{A}|$ such that $\Gamma = \Gamma_1 \Gamma_2$. Then $rank(A_h) \leq K$.*

The next proposition, which is a straightforward adaptation of a result from [49] [2], shows that the ranks of the model misfit matrices are also low in factored MDPs [23].

**Proposition 3.** *Consider a factored MDP setting where the state space is given by $\mathcal{S} = \mathcal{O}^d$ where $d \in \mathbb{N}$ and $\mathcal{O}$ is a small finite set, and the transition matrix has a factored structure with $L$ parameters. Then $rank(A_h) \leq L$.*

### 3.3   Sample Complexity

Now that we have defined our structural complexity measure, we prove sample complexity results for DREEM. We will use a slightly different definition of $\mathcal{D}(\pi, M, M')$ than the one in Section 2, in that the last action is sampled uniformly:

**Definition 3.** *(Predicted Model Disagreement Induced by Policy)*

$$\mathcal{D}(\pi, M, M', h) = \sum_{s_{h-1}} \sum_{a_{h-1}} \sum_{s_h} \Big| P_M(s_h|s_{h-1}, a_{h-1}) P_M^{\pi, h-1}(s_{h-1}) U(a_h)$$
$$- P_{M'}(s_h|s_{h-1}, a_{h-1}) P_{M'}^{\pi, h-1}(s_{h-1}) U(a_h) \Big|$$

We begin by proving a lemma which, intuitively, states that if a policy induces disagreement between two models of the environment, then it will also induce disagreement between at least one of these models and the true model. This means that by searching for and then executing a policy which induces disagreement between at least two models, the agent will collect experience from the environment which will enable it to invalidate at least one of them.

**Lemma 1.** *Let $\mathcal{M}$ be a set of models and $\Pi$ a set of policies. If there exist $M, M' \in \mathcal{M}, \pi \in \Pi$ and $h \leq H$ such that $\mathcal{D}(\pi, M, M', h) > \alpha$, then there exists $h' \leq h$ such that $\mathcal{W}(\pi, M, h') > \frac{\alpha}{4|\mathcal{A}| \cdot H}$ or $\mathcal{W}(\pi, M', h') > \frac{\alpha}{4|\mathcal{A}| \cdot H}$ (or both).*

Next, we give a lemma which states that at any time step, the agent has either found an exploration policy which will lead it to collect experience which allows it to reduce its set of candidate models, or has found an exploitation policy which is close to optimal. Here $v_\pi$ is the value of $\pi$ in the true MDP.

**Lemma 2.** *(Explore or Exploit) Suppose the true model $M^\star$ is never eliminated. At iteration $t$, one of the following two conditions must hold: either there exists $M \in \mathcal{M}_t, h_t \leq H$ such that $\mathcal{W}(\pi_{\text{explore}}^t, M, h_t) > \frac{\epsilon}{4H^2|\mathcal{A}|^2}$, or the algorithm returns $\pi_{\text{exploit}}$ such that $v_{\pi_{\text{exploit}}} > v_{\pi^\star} - \epsilon$.*

The above two lemmas state that at any time step, the agent either reduces its set of candidate models or finds an exploitation policy which is close to optimal. However, since the initial set of candidate models may be very large, we need to ensure that many models are discarded at each exploration step. Our next lemma bounds the number of iterations of the algorithm by showing that the set of candidate models is reduced by a constant factor at every step.

**Lemma 3.** *(Iteration Complexity) Let $d = \max_{1 \leq h \leq H} rank(A_h)$ and $\phi = \frac{\epsilon}{24H^2|\mathcal{A}|^2\sqrt{d}}$. Suppose that $|\widetilde{\mathcal{W}}(\pi_{\text{explore}}^t, M, h) - \mathcal{W}(\pi_{\text{explore}}^t, M, h)| \leq \phi$ holds for all $t$, $h \leq H$ and $M \in \mathcal{M}$. Then the number of rounds of Algorithm 1 with the `UpdateModelSet` routine given by Algorithm 2 is at most $Hd \log(\frac{\beta}{2\phi})/\log(5/3)$.*

The proof operates by representing each matrix $A_h$ in factored form, which induces an embedding of each model in $\mathcal{M}_t$ in a $d$-dimensional space. Minimum volume ellipsoids are then constructed around these embeddings. A geometric argument shows that the volume of these ellipsoids shrinks

by a constant factor from one iteration of the algorithm to the next, leading to a number of updates linear in $d$. Combining the previous lemmas with a concentration argument, we get our main result:

**Theorem 1.** *Assuming that* $M^\star \in \mathcal{M}$, *for any* $\epsilon, \delta \in (0,1]$ *set* $\phi = \frac{\epsilon}{24H^2|\mathcal{A}|^2\sqrt{d}}$ *and denote* $T = Hd\log(\frac{\beta}{2\phi})/\log(5/3)$. *Run Algorithm 1 with inputs* $(\mathcal{M}, n, \phi)$ *where* $n = \Theta(H^4|\mathcal{A}|^4 d\log(T|\mathcal{M}||\Pi|/\delta)/\epsilon^2)$, *and the* `UpdateModelSet` *routine is given by Algorithm 2. Then with probability at least* $1 - \delta$, *Algorithm 1 outputs a policy* $\pi_{\text{exploit}}$ *such that* $v_{\pi_{\text{exploit}}} \geq v_{\pi^\star} - \epsilon$. *The number of trajectories collected is at most* $\tilde{O}\left(\frac{H^5 d^2 |\mathcal{A}|^4}{\epsilon^2} \log\left(\frac{T|\mathcal{M}||\Pi|}{\delta}\right)\right)$.

Note that the above result requires knowledge of $d$ to set the $\phi$ and $n$ parameters. If this quantity is unknown, it can be estimated using a doubling trick which does not affect the algorithm's asymptotic sample complexity. Details can be found in Appendix A.3.

# 4 Neural-E$^3$: A Practical Instantiation

The above analysis shows that Algorithm 1 is sample efficient given an idealized instantiation, which may not be computationally practical for large model classes. Here we give a computationally efficient instantiation called Neural-E$^3$, which requires implementing the `UpdateModelSet` routine and the planning routines.

## 4.1 Model Updates

We represent $\mathcal{M}_t$ as an ensemble of action-conditional dynamics models $\{M_1, ..., M_E\}$, parameterized by neural networks, which are trained to model the next-state distribution $P_{M^\star}(s_{h+1}|s_h, a)$ using the data from the replay buffer $\mathcal{R}$. The models are trained to minimize the following loss:

$$\mathcal{L}(M, \mathcal{R}) = \mathbb{E}_{(s_{h+1}, a_h, s_h) \sim \mathcal{R}}[-\log P_M(s_{h+1}|s_h, a_h)]$$

The models in $\mathcal{M}_1$ are initialized with random weights and the subroutine `UpdateModelSet` in Algorithm 1 takes as input $\mathcal{M}_t$, performs $N_{\text{update}}$ gradient updates to each of the models using different minibatches sampled from $\mathcal{R}$, and returns the updated set of models $\mathcal{M}_{t+1}$. The dynamics models can be deterministic or stochastic (for example, Mixture Density Networks [4] or Variational Autoencoders [26]).

## 4.2 Planning

The exploration and exploitation phases require computing a policy to optimize $v_{\text{explore}}$ or $v_{\text{exploit}}$ and executing it in the environment. If the environment is deterministic, policies can be represented as action sequences, in which case we use a generalized version of breadth-first search applicable in continuous state spaces. This uses a priority queue, where expanded states are assigned a priority based on their minimum distance to other states in the currently expanded search tree. Details can be found in Appendix B.2.1. For stochastic environments, we used implicit policies obtained using Monte-Carlo Tree Search (MCTS) [10], where each node in the tree consists of empirical distributions predicted by the different models conditioned on the action sequence leading to the node. The agent only executes the first action of the sequence returned by the planning procedure, and replans at every step to account for the stochasticity of the environment. See Appendix B.2.2 for details.

## 4.3 Exploitation with Off-Policy RL

For some problems with sparse rewards, it may be computationally impractical to use planning during the exploitation phase, even with a perfect model. Note that much of the exploration phase, which uses model disagreement as a fictitious reward, can be seen as an MDP with dense rewards, while the rewards in the true MDP may be sparse. In these settings, we use an alternative approach where a parameterized value function such as a DQN [31] is trained using the experience collected in the replay buffer during exploration. This can be done offline without collecting additional samples from the environment. We also found this useful for problems with antishaped rewards, where the MCTS procedure can be biased away from the optimal actions if they temporarily lead to lower reward than suboptimal ones.

## 4.4 Relationship between Idealized and Practical Algorithms

For both the idealized and practical algorithms, $\mathcal{M}_t$ represents a set of models with low error on the current replay buffer. In the idealized algorithm, models with high error are eliminated explicitly in Algorithm 2, while in the practical algorithm, models with high error are avoided by the optimization procedure. The main difference between the two algorithms is that the idealized version maintains *all* models in the model class which have low error (which includes the true model), whereas the practical version only maintains a subset due to time and memory constraints. A potential failure mode of the practical algorithm would be if all the models wrongly agree in their predictions in some unexplored part of the state-action space which leads to high reward. However, in practice we found that using different initializations and minibatches was sufficient to obtain a diverse set of models, and that using even a relatively small ensemble (4 to 8 models) led to successful exploration.

## 5 Related Work

Theoretical guarantees for a number of model-based RL algorithms exist in the tabular setting [24, 5, 47, 45] and in the continuous setting when the dynamics are assumed to be linear [51, 1, 11]. The Metric-$E^3$ algorithm [22] operates in general state spaces, but its sample complexity depends on the covering number which may be exponential in dimension. The algorithm of [29] addresses general model classes and optimizes lower bounds on the value function, and provably converges to a *locally* optimal policy with a number of samples polynomial in the dimension of the state space. It also admits an approximate instantiation which was shown to work well in continuous control tasks. The work of [49] provides an algorithm which provably recovers a *globally* near-optimal policy with polynomial sample complexity using a structural complexity measure which we adapt for our analysis, but does not investigate practical approximations. The algorithm we analyze is fundamentally different from both of these approaches, as it uses disagreement over predicted states rather than optimism to drive exploration.

Our practical approximation is closely related to the MAX algorithm [44], which also uses disagreement between different models in an ensemble to drive exploration. Our version differs in a few ways i) we use maximal disagreement rather than variance to measure uncertainty, as this reflects our theoretical analysis ii) we define the exploration MDP differently, by propagating the state predictions of the different models rather than sampling at each step iii) we explicitly address the exploitation step, whereas they focused primarily on exploration. The work of [40] also used disagreement between single-step predictions to train an exploration policy.

Several works have empirically demonstrated the sample efficiency of model-based RL in continuous settings [2, 12, 46, 34, 8], including with high-dimensional images [17, 19, 16]. These have primarily focused on settings with dense rewards where simple exploration was sufficient, or where rich observational data was available.

Other approaches to exploration include augmenting rewards with exploration bonuses, such as inverse counts in the tabular setting [48, 27], pseudo-counts derived from density models over the state space [3, 38], prediction errors of either a dynamics model [39] or a randomly initialized network [7], or randomizing value functions [36, 37]. These have primarily focused on model-free methods, which have been known to have high sample complexity despite yielding good final performance.

## 6 Experiments

We now give empirical results for the Neural-$E^3$ algorithm described in Section 4. See Appendix C for experimental details and https://github.com/mbhenaff/neural-e3 for source code.

### 6.1 Stochastic Combination Lock

We begin with a set of experiments on the stochastic combination lock environment described in [14] and shown in Figure 1(a). These environments consist of $H$ levels with 3 states per level and 4 actions. Two of the states lead to high reward and the third is a dead state from which it is impossible to recover. The effect of actions are flipped with probability 0.1, and the one-hot state encodings are appended with random Bernoulli noise to increase the number of possible observations. We

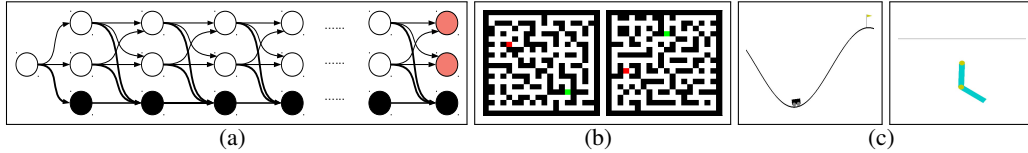
(a)                          (b)                       (c)

Figure 1: Environments tested. a) **Stochastic combination lock:** The agent must reach the red states to collect high reward while avoiding the dead states (black) from which it cannot recover. b) **Mazes:** The agent (green) must navigate through the maze to reach the goal (red). Different mazes are generated each episode, which requires generalizing across mazes (colors are changed here for readability) c) **Continuous Control**: Classic control tasks requiring non-trivial exploration.

experimented with two task variants: a first where the rewards are zero everywhere except for the red states, and a second where small, antishaped rewards encourage the agent to transition to the dead states (see Appendix C.1 for details). This tests an algorithm's robustness to poor local optima.

We compare against three other methods: a double DQN [18] with prioritized experience replay [41] using the OpenAI Baselines implementation [13], a Proximal Policy Optimization (PPO) agent [42], and a PPO agent with a Random Network Distillation (RND) exploration bonus [7]. For Neural-E$^3$, we used stochastic dynamics models outputting the parameters of a multivariate Bernoulli distribution, and the MCTS procedure described in Appendix B.2.2 during the exploration phase. We used the DQN-based method described in Section 4.3 for the exploit phase.

Figure 2(a) shows performance across 5 random seeds for the first variant of the task. For all horizons, Neural-E$^3$ achieves the optimal reward across most seeds. The DQN also performs well, although it often requires more samples than Neural-E$^3$. For longer horizons, PPO never collects rewards, while PPO+RND eventually succeeds given a large number of episodes (see Appendix C.1).

Figure 2(b) shows results for the task variant with antishaped rewards. For longer horizons, Neural-E$^3$ is the only method to achieve the globally optimal reward, whereas none of the other methods get past the poor local optimum induced by the misleading rewards. Note that Neural-E$^3$ actually obtains *less* reward than the other methods during its exploration phase, but this pays off during exploitation since it enables the agent to eventually discover states with much higher reward.

## 6.2 Maze Environment

We next evaluated our approach on a maze environment, which is a modified version of the Collect domain [35], shown in Figure 1(b). States consist of RGB images where the three channels represent the walls, the agent and the goal respectively. The agent receives a reward of $2.0$ for reaching the goal, $-0.5$ for hitting a wall and $-0.2$ otherwise. Mazes are generated randomly for each episode, thus the number of states is extremely large and the agent must learn to generalize across mazes. Our dynamics models are action-conditional convolutional networks taking as input an image and action and predicting the next image and reward. We used the deterministic search procedure described in Section B.2.1 for planning.

We compared to two other approaches. The first was a double DQN with prioritized experience replay as before. The second was a model-based agent identical to ours, except that it uses a uniform exploration policy during the explore phase. This is similar to the PETS algorithm [8] applied to discrete action spaces, as it optimizes rewards over an ensemble of dynamics models. We call this UE$^2$, for Uniform Explore Exploit.

Performance measured by reward across 3 random seeds is shown in Figure 2(c) for different maze sizes. The DQN agent is able to solve the smallest $5 \times 5$ mazes after a large number of episodes, but is not able to learn meaningful behavior for larger mazes. The UE$^2$ and Neural-E$^3$ agents both perform similarly for the $5 \times 5$ mazes, but the relative performance of Neural-E$^3$ improves as the size of the maze becomes larger. Note also that the Neural-E$^3$ agent collects more reward during its exploration phase, even though it is not explicitly optimizing for reward but rather for model disagreement. Figure 5 in Appendix C.2 shows the model predictions for an action sequence executed by the Neural-E$^3$ agent during the exploration phase. The predictions of the different models agree until the reward is reached, which is a rare event.

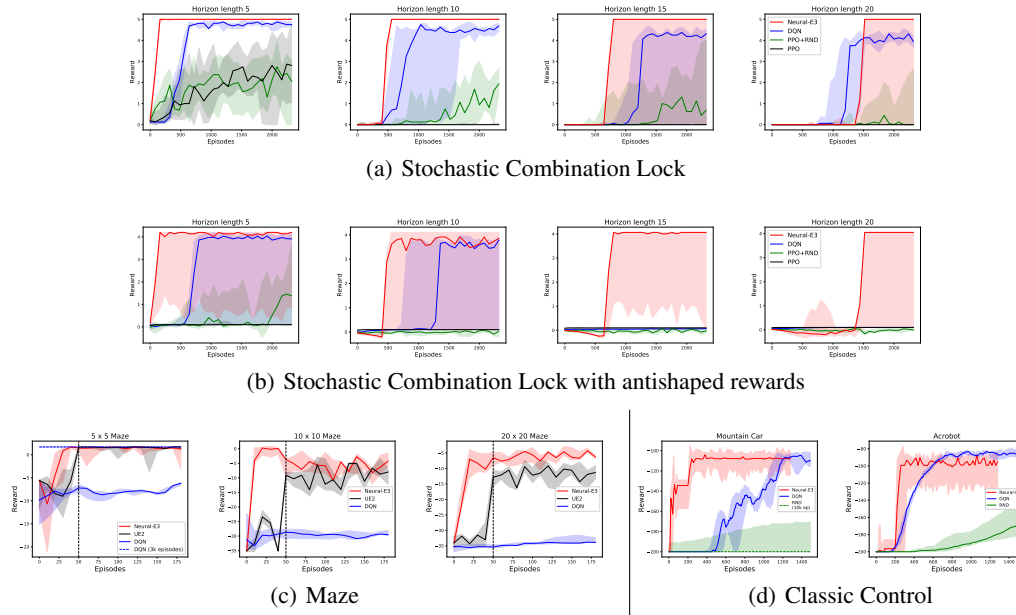

(a) Stochastic Combination Lock

(b) Stochastic Combination Lock with antishaped rewards

(c) Maze                                   (d) Classic Control

Figure 2: Comparison of methods across different domains. Solid lines represent median performance across seeds, shaded region represents range between best and worst seeds.

## 6.3 Continuous Control

We then evaluated our approach on two continuous control domains, shown in Figure 1(c). Mountain-Car [32] is an environment with simple non-linear dynamics and continuous state space ($\mathcal{S} \subseteq \mathbb{R}^2$) where the agent must drive an underpowered car up a steep hill, which requires building momentum by first driving up the opposite end of the hill. The agent only receives reward at the top of the hill, hence this requires non-trivial exploration. Acrobot [50] requires swinging a simple under-actuated robot above a given height, also with a continuous state space ($\mathcal{S} \subseteq \mathbb{R}^6$). Both tasks have discrete action spaces with $|\mathcal{A}| = 3$.

We found that even planning with a perfect model was computationally impractical due to the sparsity of rewards, hence we used the method described in section 4.3, where we trained a DQN offline using the data collected during exploration. Results for Neural-$E^3$, DQN and RND agents across 5 random seeds are shown in Figure 2(d). For Mountain Car, the DQN is able to solve the task but requires around 1200 episodes to do so. Neural-$E^3$ is able to quickly explore, and solves the task to a similar degree of success in under 300 episodes. The RND agent only starts to collect reward after 10K episodes. For the Acrobot task, Neural-$E^3$ also explores quickly, although its increase in sample efficiency is less pronounced compared to the DQN. The RND agent is also able to make quicker progress on this task, which suggests that the exploration problem may not be as difficult.

## 7 Conclusion

This work extends the classic $E^3$ algorithm to operate in large or infinite state spaces. On the theoretical side, we present a model-elimination based version of the algorithm which provably requires only a polynomial number of samples to learn a near-optimal policy with high probability. Empirically, we show that this algorithm can be approximated using neural networks and still provide good sample efficiency in practice. An interesting direction for future work would be combining the exploration and exploitation phases in a unified process, which has been done in the tabular setting [5]. Another direction would be to explicitly encourage disagreement between different models in the ensemble for unseen inputs, in order to better approximate the maximal disagreement between models in a version space which we use in our idealized algorithm. Such ideas have been proposed in active learning [9] and contextual bandits [15], and could potentially be adapted to multi-step RL.

**Acknowledgments**

I would like to thank Akshay Krishnamurthy, John Langford, Alekh Agarwal and Miro Dudik for helpful discussions and feedback.

## Footnotes

[1] We use a generalized notion of rank with a condition on the row norms of the factorization: for an $m \times n$ matrix $B$, denote $rank(B, \beta)$ to be the smallest integer $k$ such that $B = UV^\top$ with $U \in \mathbb{R}^{m \times k}, V \in \mathbb{R}^{n \times k}$ and for every pair of rows $u_i, v_j$ we have $\|u_i\|_2 \cdot \|v_j\|_2 \leq \beta$. $\beta$ appears in Lemma 3 and Theorem 1.

[2] Appendix E.2, Proposition 2

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
