[Supplementary Material]

# A Proofs

## A.1 Computing Empirical Misfit

Estimating the misfit $\mathcal{W}(\pi, M, h)$ directly may not be possible when dealing with large state spaces, since a given roll-in state-action pair $(s_{h-1}, a_{h-1})$ may only be observed once and we do not have access to the true model to compute the distribution $P_{M^\star}(\cdot|s_{h-1}, a_{h-1})$. However, we can use an alternate approach based on Integral Probability Metrics (IPMs) [33] similar to that described in Appendix B of [49]. Let $\mathcal{F} = \{f : \mathcal{S} \times \mathcal{A} \times \mathcal{S} \to \mathbb{R} : \|f\|_\infty \leq 1\}$. Using this class of test functions, the total variation distance can be written as:

$$\|P_M(\cdot|s_{h-1}, a_{h-1}) - P_{M^\star}(\cdot|s_{h-1}, a_{h-1})\|_{TV} =$$
$$\max_{f \in \mathcal{F}} \underbrace{\mathbb{E}_{s_h \sim P_M(\cdot|s_{h-1}, a_{h-1})}[f(s_{h-1}, a_{h-1}, s_h)] - \mathbb{E}_{s_h \sim P_{M^\star}(\cdot|s_{h-1}, a_{h-1})}[f(s_{h-1}, a_{h-1}, s_h)]}_{g(M, f, s_{h-1}, a_{h-1})}$$

The next lemma shows that $\mathcal{W}(\pi, M, h)$ can be expressed using the IPM definition with the max operator placed outside both expectation operators. This will then allow us to estimate the misfit using a smaller (finite) set of test functions and apply concentration arguments to bound the difference between the true and estimated values.

**Technical Lemma 1.**

$$\mathcal{W}(\pi, M, h) =$$
$$\max_{f \in \mathcal{F}} \mathbb{E}_{s_{h-1} \sim P_{M^\star}^{\pi, h-1}, a_{h-1} \sim U(\mathcal{A})} \Big[ \mathbb{E}_{s_h \sim P_M(\cdot|s_{h-1}, a_{h-1})}[f(s_{h-1}, a_{h-1}, s_h)] - \mathbb{E}_{s_h \sim P_{M^\star}(\cdot|s_{h-1}, a_{h-1})}[f(s_{h-1}, a_{h-1}, s_h)] \Big]$$

*Proof.* Define $f_{s,a,M}^{\max} = \operatorname{argmax}_{f \in \mathcal{F}} g(M, f, s, a)$ and $f_M^{\max} : \mathcal{S} \times \mathcal{A} \times \mathcal{S} \to \mathbb{R}$ by $f_M^{\max}(s, a, s') = f_{s,a,M}^{\max}(s, a, s')$. Note that $\|f_M^{\max}\|_\infty \leq 1$ so $f_M^{\max} \in \mathcal{F}$.

We can then write:

$$\mathcal{W}(\pi, M, h) = \mathbb{E}_{s_{h-1} \sim P_{M^\star}^{\pi, h-1}, a_{h-1} \sim U(\mathcal{A})} \Big[ \max_{f \in \mathcal{F}} g(M, f, s_{h-1}, a_{h-1}) \Big]$$
$$= \mathbb{E}_{s_{h-1} \sim P_{M^\star}^{\pi, h-1}, a_{h-1} \sim U(\mathcal{A})} \Big[ g(M, f_{s_{h-1}, a_{h-1}, M}^{\max}, s_{h-1}, a_{h-1}) \Big]$$
$$= \mathbb{E}_{s_{h-1} \sim P_{M^\star}^{\pi, h-1}, a_{h-1} \sim U(\mathcal{A})} \Big[ g(M, f_M^{\max}, s_{h-1}, a_{h-1}) \Big]$$
$$\leq \max_{f \in \mathcal{F}} \mathbb{E}_{s_{h-1} \sim P_{M^\star}^{\pi, h-1}, a_{h-1} \sim U(\mathcal{A})} \Big[ g(M, f, s_{h-1}, a_{h-1}) \Big]$$

Now let $f^\star$ be the function which maximizes the last quantity. We then have:

$$\max_{f \in \mathcal{F}} \mathbb{E}_{s_{h-1} \sim P_{M^\star}^{\pi, h-1}, a_{h-1} \sim U(\mathcal{A})} \Big[ g(M, f, s_{h-1}, a_{h-1}) \Big]$$
$$= \mathbb{E}_{s_{h-1} \sim P_{M^\star}^{\pi, h-1}, a_{h-1} \sim U(\mathcal{A})} \Big[ g(M, f^\star, s_{h-1}, a_{h-1}) \Big]$$
$$\leq \mathbb{E}_{s_{h-1} \sim P_{M^\star}^{\pi, h-1}, a_{h-1} \sim U(\mathcal{A})} \Big[ \max_{f \in \mathcal{F}} g(M, f, s_{h-1}, a_{h-1}) \Big]$$
$$= \mathcal{W}(\pi, M, h)$$

Combining the two inequalities gives the result.

$\square$

We next define a new set of functions $\widetilde{\mathcal{F}}$ as follows. Let

$$f_{\pi, M, h} = \operatorname{argmax}_{f \in \mathcal{F}} \mathbb{E}_{s_{h-1} \sim P_{M^\star}^{\pi, h-1}, a_{h-1} \sim U(\mathcal{A})} \Big[ g(M, f, s_{h-1}, a_{h-1}) \Big]$$

and $\widetilde{\mathcal{F}} = \{\pm f_{\pi,M,h} : \pi \in \Pi, M \in \mathcal{M}, h \in [H]\}$. We then have:

$$\mathcal{W}(\pi, M, h) = \max_{f \in \mathcal{F}} \mathbb{E}_{s_{h-1} \sim P_{M^\star}^{\pi,h-1}, a_{h-1} \sim U(\mathcal{A})} \Big[ g(M, f, s_{h-1}, a_{h-1}) \Big]$$

$$= \max_{f \in \widetilde{\mathcal{F}}} \mathbb{E}_{s_{h-1} \sim P_{M^\star}^{\pi,h-1}, a_{h-1} \sim U(\mathcal{A})} \Big[ g(M, f, s_{h-1}, a_{h-1}) \Big]$$

The misfit can thus be computed using a smaller (finite) set of test functions $\widetilde{\mathcal{F}}$, with size $|\widetilde{\mathcal{F}}| \leq |\Pi| \cdot |\mathcal{M}| \cdot H$.

Given a dataset $\mathcal{R}_\pi = \{(s_{h-1}^{(i)}, a_{h-1}^{(i)}, s_h^{(i)})\}_{n=1}^n$ generated by following policy $\pi$, we estimate the empirical misfit for a model $M$ at time step $h$ using $\widetilde{\mathcal{F}}$ as follows:

$$\widetilde{\mathcal{W}}(\pi, M, h) = \max_{f \in \widetilde{\mathcal{F}}} \frac{1}{n} \sum_{i=1}^n \Big[ \mathbb{E}_{s_h \sim P_M(\cdot|s_{h-1}, a_{h-1})}[f(s_{h-1}, a_{h-1}, s_h)] - f(s_{h-1}^{(i)}, a_{h-1}^{(i)}, s_h^{(i)}) \Big]$$

**Technical Lemma 2.** *(Deviation Bound for $\widetilde{\mathcal{W}}(\pi, M, h)$). Fix $h$ and $\pi \in \Pi$. Sample a dataset $\left\{(s_{h-1}^{(i)}, a_{h-1}^{(i)}, s_h^{(i)})\right\}_{i=1}^n$ of size $n$ with:*

$$s_{h-1}^{(i)} \sim P_{M^\star}^{\pi,h-1}, a_{h-1}^{(i)} \sim U(\mathcal{A}), s_h^{(i)} \sim P_{M^\star}(\cdot|s_{h-1}^{(i)}, a_{h-1}^{(i)})$$

*Then with probability at least $1 - \delta$, we have for all $M \in \mathcal{M}$:*

$$\left| \widetilde{\mathcal{W}}(\pi, M, h) - \mathcal{W}(\pi, M, h) \right| \leq \frac{4 \log(2|\mathcal{M}||\Pi|H/\delta)}{3n} + 4\sqrt{\frac{2 \log(2|\mathcal{M}||\Pi|H/\delta)}{n}}$$

*Proof.* Fix $M \in \mathcal{M}$ and $f \in \widetilde{\mathcal{F}}$. Define the random variable $z_i(M, f)$ as

$$z_i(M, f) = \mathbb{E}_{s_h \sim P_M(\cdot|s_{h-1}^{(i)}, a_{h-1}^{(i)})} f(s_{h-1}^{(i)}, a_{h-1}^{(i)}, s_h) - f(s_{h-1}^{(i)}, a_{h-1}^{(i)}, s_h^{(i)})$$

The expectation is given by:

$$\mathbb{E}[z_i(M, f)] = \mathbb{E}_{s_{h-1} \sim P_{M^\star}^{\pi,h}, a_{h-1} \sim U} \Big[ \mathbb{E}_{s_h \sim P_M(\cdot|s_{h-1}, a_{h-1})}[f(s_{h-1}, a_{h-1}, s_h)] - \mathbb{E}_{s_h \sim P_{M^\star}(\cdot|s_{h-1}, a_{h-1})}[f(s_{h-1}, a_{h-1}, s_h)] \Big]$$

Note that $|z_i(M, f)| \leq 2$ and $\text{Var}(z_i(M, f)) \leq 2$. Therefore we can apply Bernstein's inequality which states that for any $\epsilon$:

$$P\Big[ \Big| \sum_{i=1}^n (z_i(M, f) - \mathbb{E}[z_i(M, f)]) \Big| > \epsilon \Big] \leq 2\exp\Big( -\frac{\epsilon^2/2}{\sum_{i=1}^n \mathbb{E}[(z_i(M, f) - \mathbb{E}[z_i(M, f)])^2] + 2\epsilon/3} \Big)$$

$$\leq 2\exp\Big( -\frac{\epsilon^2/2}{2n + 2\epsilon/3} \Big) \triangleq \delta$$

Solving for $\epsilon$ in terms of $\delta$, we get: $\frac{\epsilon^2/2}{2n+2\epsilon/3} = \log(2/\delta) \implies \epsilon^2 - 4n \log(2/\delta) - \frac{2}{3}\epsilon \log(2/\delta) = 0$.
Applying the quadratic formula then gives us:

$$\epsilon = \frac{1}{3}\log(2/\delta) + \frac{1}{2}\sqrt{(\frac{2}{3}\log(2/\delta))^2 + 16n\log(2/\delta)}$$

$$\leq \frac{1}{3}\log(2/\delta) + \frac{1}{2}\sqrt{(\frac{2}{3}\log(2/\delta))^2 + \sqrt{16n\log(2/\delta)}}$$

$$= \frac{2}{3}\log(2/\delta) + 4\sqrt{n\log(2/\delta)}$$

Therefore with probability at least $1 - \delta$ we have:

$$\left|\sum_{i=1}^{n}(z_i(M,f) - \mathbb{E}[z_i(M,f)])\right| < \epsilon \leq \frac{2}{3}\log(2/\delta) + 4\sqrt{n\log(2/\delta)}$$

And therefore:

$$\left|\left[\frac{1}{n}\sum_{i=1}^{n}(z_i(M,f)\right] - \mathbb{E}[z_i(M,f)]\right| \leq \frac{2\log(2/\delta)}{3n} + 4\frac{\sqrt{n\log(2/\delta)}}{n} = \frac{2\log(2/\delta)}{3n} + 4\sqrt{\frac{\log(2/\delta)}{n}}$$

Via a union bound over $\mathcal{M}$ and $\widetilde{\mathcal{F}}$, we have that for all pairs $M \in \mathcal{F}$ and $f \in \widetilde{\mathcal{F}}$, with probability at least $1 - \delta$:

$$\left|\left[\frac{1}{n}\sum_{i=1}^{n}(z_i(M,f)\right] - \mathbb{E}[z_i(M,f)]\right| \leq \frac{2\log(2|\mathcal{M}||\widetilde{\mathcal{F}}|/\delta)}{3n} + 4\sqrt{\frac{\log(2|\mathcal{M}||\widetilde{\mathcal{F}}|/\delta)}{n}}$$

$$\leq \frac{2\log(2|\mathcal{M}|^2|\Pi|H/\delta)}{3n} + 4\sqrt{\frac{\log(2|\mathcal{M}|^2|\Pi|H/\delta)}{n}}$$

$$\leq \frac{4\log(2|\mathcal{M}||\Pi|H/\delta)}{3n} + 4\sqrt{\frac{2\log(2|\mathcal{M}||\Pi|H/\delta)}{n}}$$

Note that $\mathcal{W}(\pi, M, h) = \max_{f \in \widetilde{\mathcal{F}}} \mathbb{E}[z_i(M,f)]$ and $\widetilde{\mathcal{W}}(\pi, M, h) = \max_{f \in \widetilde{\mathcal{F}}} \frac{1}{n}\sum_{i=1}^{n} z_i(M,f)$. For a fixed $M$, we have shown uniform convergence over $\widetilde{\mathcal{F}}$ which implies that the empirical and population maxima must be similarly close, which yields the result.

$\square$

## A.2 Main Results

**Proposition 1.** *Assume $|\mathcal{S}|$ is finite and let $A_h$ be the matrix defined above. Then $rank(A_h) \leq |\mathcal{S}|$.*

*Proof.* This proposition is a special case of Proposition 2 so the proof carries over. It can also be shown with a direct argument as follows. Define the matrix $U_h \in \mathbb{R}^{|\Pi| \times |\mathcal{S}|}$ by $U_h(\pi, s) = P_{M^\star}^{\pi, h-1}(s)$ and the matrix $V_h \in \mathbb{R}^{|\mathcal{M}| \times |\mathcal{S}|}$ by $V_h(M, s) = \mathbb{E}_{a \sim U(\mathcal{A})}[\|P_M(\cdot|s,a) - P_{M^\star}(\cdot|s,a)\|_{TV}]$.

Then we can write:

$$A_h(\pi, M) = \mathcal{W}(\pi, M, h) = \mathbb{E}_{s \sim P_{M^\star}^{\pi,h-1}, a \sim U(\mathcal{A})} \left[ \| P_M(\cdot|s,a) - P_{M^\star}(\cdot|s,a) \|_{TV} \right]$$

$$= \sum_s P_{M^\star}^{\pi,h-1}(s) \mathbb{E}_{a \sim U(\mathcal{A})} \left[ \| P_M(\cdot|s,a) - P_{M^\star}(\cdot|s,a) \|_{TV} \right]$$

$$= \sum_s U_h(\pi, s) V_h(M, s)$$

Therefore we have $A_h = U_h V_h^\top$ and so $rank(A_h) \le |\mathcal{S}|$. □

**Proposition 2.** *Let $\Gamma$ denote the true transition matrix of size $|\mathcal{S}| \times |\mathcal{S} \times \mathcal{A}|$, with $\Gamma(s', (s,a)) = P_{M^\star}(s'|s,a)$. Assume that there exist two matrices $\Gamma_1, \Gamma_2$ with sizes $|\mathcal{S}| \times K$ and $K \times |\mathcal{S} \times \mathcal{A}|$ such that $\Gamma = \Gamma_1 \Gamma_2$. Then $rank(A_h) \le K$.*

*Proof.* We first define the vectors $z^{\pi,h}$ of size $K$ as follows:

$$z_k^{\pi,h} = \sum_{s_{h-1}} \sum_{a_{h-1}} P_{M^\star}^{\pi,h-1}(s_{h-1}) \pi(a_{h-1}|s_{h-1}) \Gamma_2(k, (s_{h-1}, a_{h-1}))$$

This allows us to rewrite:

$$P_{M^\star}^{\pi,h}(s_h) = \sum_{s_{h-1}} \sum_{a_{h-1}} P_{M^\star}^{\pi,h-1}(s_{h-1}) \pi(a_{h-1}|s_{h-1}) P_{M^\star}(s_h|s_{h-1}, a_{h-1})$$

$$= \sum_{s_{h-1}} \sum_{a_{h-1}} \sum_{k=1}^K P_{M^\star}^{\pi,h-1}(s_{h-1}) \pi(a_{h-1}|s_{h-1}) \Gamma_2(k, (s_{h-1}, a_{h-1})) \Gamma_1(s_h, k)$$

$$= \sum_{k=1}^K z_k^{\pi,h} \Gamma_1(s_h, k)$$

We can now rewrite the witnessed model misfit as follows:

$$\mathcal{W}(\pi, M, h) = \mathbb{E}_{s_{h-1} \sim P_{M^\star}^{\pi,h-1}, a_{h-1} \sim U(\mathcal{A})} \left[ \| P_M(\cdot|s_{h-1}, a_{h-1}) - P_{M^\star}(\cdot|s_{h-1}, a_{h-1}) \|_{TV} \right]$$

$$= \sum_{s_{h-1}} \sum_{a_{h-1}} P_{M^\star}^{\pi,h-1}(s_{h-1}) \frac{1}{|\mathcal{A}|} \left[ \| P_M(\cdot|s_{h-1}, a_{h-1}) - P_{M^\star}(\cdot|s_{h-1}, a_{h-1}) \|_{TV} \right]$$

$$= \sum_{s_{h-1}} \sum_{a_{h-1}} \sum_{k=1}^K z_k^{\pi,h-1} \Gamma_1(s_{h-1}, k) \frac{1}{|\mathcal{A}|} \left[ \| P_M(\cdot|s_{h-1}, a_{h-1}) - P_{M^\star}(\cdot|s_{h-1}, a_{h-1}) \|_{TV} \right]$$

$$= \sum_{k=1}^K z_k^{\pi,h-1} \sum_{s_{h-1}} \sum_{a_{h-1}} \Gamma_1(s_{h-1}, k) \frac{1}{|\mathcal{A}|} \left[ \| P_M(\cdot|s_{h-1}, a_{h-1}) - P_{M^\star}(\cdot|s_{h-1}, a_{h-1}) \|_{TV} \right]$$

Define the matrices $U_h$ and $V_h$ of size $|\Pi| \times K$ and $|\mathcal{M}| \times K$ by:

$$U_h(\pi, k) = z_k^{\pi,h-1}$$

$$V_h(M, k) = \sum_{s_{h-1}} \sum_{a_{h-1}} \Gamma_1(s_{h-1}, k) \frac{1}{|\mathcal{A}|} \left[ \| P_M(\cdot|s_{h-1}, a_{h-1}) - P_{M^\star}(\cdot|s_{h-1}, a_{h-1}) \|_{TV} \right]$$

We then have $A_h = U_h V_h^\top$, which proves the desired result. □

**Lemma 1.** *Let $\mathcal{M}$ be a set of models and $\Pi$ a set of policies. If there exist $M, M' \in \mathcal{M}, \pi \in \Pi$ and $h \leq H$ such that $\mathcal{D}(\pi, M, M', h) > \alpha$, then there exists $h' \leq h$ such that $\mathcal{W}(\pi, M, h') > \frac{\alpha}{4|\mathcal{A}| \cdot H}$ or $\mathcal{W}(\pi, M', h') > \frac{\alpha}{4|\mathcal{A}| \cdot H}$ (or both).*

*Proof.* If there exists $h' \leq h-1$ such that either $\mathcal{W}(\pi, M, h') > \frac{\alpha}{4|\mathcal{A}| \cdot H}$ or $\mathcal{W}(\pi, M', h') > \frac{\alpha}{4|\mathcal{A}| \cdot H}$ then we are done. Therefore assume that $\mathcal{W}(\pi, M, h'), \mathcal{W}(\pi, M', h') \leq \frac{\alpha}{4|\mathcal{A}| \cdot H}$ for all $h' \in [H-1]$. To keep notation light, for the following we will use the following abbreviations:

$$P_M^h := P_M(s_h | s_{h-1}, a_{h-1})$$
$$P_M^{\pi, h-1} := P_M^{\pi, h-1}(s_{h-1})$$

We now write:

$\mathcal{D}(\pi, M, M', h)$

$= \dfrac{1}{|\mathcal{A}|} \displaystyle\sum_{s_{h-1}} \sum_{a_{h-1}} \sum_{s_h} |P_M^h P_M^{\pi, h-1} - P_{M'}^h P_{M'}^{\pi, h-1}|$

$= \dfrac{1}{|\mathcal{A}|} \displaystyle\sum_{s_{h-1}} \sum_{a_{h-1}} \sum_{s_h} |P_M^h P_M^{\pi, h-1} - P_{M'}^h P_{M'}^{\pi, h-1} - P_M^h P_{M^\star}^{\pi, h-1} + P_M^h P_{M^\star}^{\pi, h-1} + P_{M'}^h P_{M^\star}^{\pi, h-1} - P_{M'}^h P_{M^\star}^{\pi, h-1}|$

$= \dfrac{1}{|\mathcal{A}|} \displaystyle\sum_{s_{h-1}} \sum_{a_{h-1}} \sum_{s_h} |P_M^h(P_M^{\pi, h-1} - P_{M^\star}^{\pi, h-1}) + P_{M'}^h(P_{M^\star}^{\pi, h-1} - P_{M'}^{\pi, h-1}) + (P_M^h - P_{M'}^h)P_{M^\star}^{\pi, h-1}|$

$\leq \dfrac{1}{|\mathcal{A}|} \displaystyle\sum_{s_{h-1}} \sum_{a_{h-1}} \sum_{s_h} P_M^h|P_M^{\pi, h-1} - P_{M^\star}^{\pi, h-1}| + P_{M'}^h|P_{M^\star}^{\pi, h-1} - P_{M'}^{\pi, h-1}| + |P_M^h - P_{M'}^h|P_{M^\star}^{\pi, h-1}$

$\leq \dfrac{1}{|\mathcal{A}|} \displaystyle\sum_{s_{h-1}} \sum_{a_{h-1}} \sum_{s_h} P_M^h|P_M^{\pi, h-1} - P_{M^\star}^{\pi, h-1}| + P_{M'}^h|P_{M^\star}^{\pi, h-1} - P_{M'}^{\pi, h-1}| + |P_M^h - P_{M^\star}^h|P_{M^\star}^{\pi, h-1} + |P_{M'}^h - P_{M^\star}^h|P_{M^\star}^{\pi, h-1}$

$= \dfrac{1}{|\mathcal{A}|} \displaystyle\sum_{s_{h-1}} \sum_{a_{h-1}} \sum_{s_h} P_M^h|P_M^{\pi, h-1} - P_{M^\star}^{\pi, h-1}| + \dfrac{1}{|\mathcal{A}|} \displaystyle\sum_{s_{h-1}} \sum_{a_{h-1}} \sum_{s_h} P_{M'}^h|P_{M^\star}^{\pi, h-1} - P_{M'}^{\pi, h-1}|$

$\qquad + \dfrac{1}{|\mathcal{A}|} \displaystyle\sum_{s_{h-1}} \sum_{a_{h-1}} \sum_{s_h} |P_M^h - P_{M^\star}^h|P_{M^\star}^{\pi, h-1} + \dfrac{1}{|\mathcal{A}|} \displaystyle\sum_{s_{h-1}} \sum_{a_{h-1}} \sum_{s_h} |P_{M'}^h - P_{M^\star}^h|P_{M^\star}^{\pi, h-1}$

$= \dfrac{1}{|\mathcal{A}|} \displaystyle\sum_{s_{h-1}} \sum_{a_{h-1}} \sum_{s_h} P_M^h|P_M^{\pi, h-1} - P_{M^\star}^{\pi, h-1}| + \dfrac{1}{|\mathcal{A}|} \displaystyle\sum_{s_{h-1}} \sum_{a_{h-1}} \sum_{s_h} P_{M'}^h|P_{M^\star}^{\pi, h-1} - P_{M'}^{\pi, h-1}| + \mathcal{W}(\pi, M, h) + \mathcal{W}(\pi, M', h)$

We now bound the first term in this sum:

$$\frac{1}{|\mathcal{A}|} \sum_{s_{h-1}} \sum_{a_{h-1}} \sum_{s_h} P_M^h \left| P_M^{\pi,h-1} - P_{M^\star}^{\pi,h-1} \right|$$

$$= \sum_{s_{h-1}} \left| P_M^{\pi,h-1} - P_{M^\star}^{\pi,h-1} \right|$$

$$= \sum_{s_{h-1}} \left| \sum_{s_{h-2}} \sum_{a_{h-2}} \pi(a_{h-2}|s_{h-2})(P_M^{h-1} P_M^{\pi,h-2} - P_{M^\star}^{h-1} P_{M^\star}^{\pi,h-2}) \right|$$

$$= \sum_{s_{h-1}} \sum_{s_{h-2}} \sum_{a_{h-2}} \pi(a_{h-2}|s_{h-2}) \left| P_M^{h-1} P_M^{\pi,h-2} - P_{M^\star}^{h-1} P_{M^\star}^{\pi,h-2} - P_M^{h-1} P_{M^\star}^{\pi,h-2} + P_M^{h-1} P_{M^\star}^{\pi,h-2} \right|$$

$$= \sum_{s_{h-1}} \sum_{s_{h-2}} \sum_{a_{h-2}} \pi(a_{h-2}|s_{h-2}) \left| P_M^{h-1}(P_M^{\pi,h-2} - P_{M^\star}^{\pi,h-2}) + (P_M^{h-1} - P_{M^\star}^{h-1}) P_{M^\star}^{\pi,h-2} \right|$$

$$\leq \sum_{s_{h-1}} \sum_{s_{h-2}} \sum_{a_{h-2}} \pi(a_{h-2}|s_{h-2}) \left| P_M^{h-1}(P_M^{\pi,h-2} - P_{M^\star}^{\pi,h-2}) \right| + \sum_{s_{h-1}} \sum_{s_{h-2}} \sum_{a_{h-2}} \pi(a_{h-2}|s_{h-2}) \left| (P_M^{h-1} - P_{M^\star}^{h-1}) P_{M^\star}^{\pi,h-2} \right|$$

$$\leq \sum_{s_{h-1}} \sum_{s_{h-2}} \sum_{a_{h-2}} \pi(a_{h-2}|s_{h-2}) \left| P_M^{h-1}(P_M^{\pi,h-2} - P_{M^\star}^{\pi,h-2}) \right| + \sum_{s_{h-1}} \sum_{s_{h-2}} \sum_{a_{h-2}} \left| (P_M^{h-1} - P_{M^\star}^{h-1}) P_{M^\star}^{\pi,h-2} \right|$$

$$= \sum_{s_{h-2}} \left[ \sum_{s_{h-1}} \left[ \sum_{a_{h-2}} \pi(a_{h-2}|s_{h-2}) \right] P_M^{h-1} \right] \left| P_M^{\pi,h-2} - P_{M^\star}^{\pi,h-2} \right| + |\mathcal{A}| \cdot \mathcal{W}(\pi, M, h-1)$$

$$= \sum_{s_{h-2}} \left| P_M^{\pi,h-2} - P_{M^\star}^{\pi,h-2} \right| + |\mathcal{A}| \cdot \mathcal{W}(\pi, M, h-1)$$

$$\leq \sum_{s_{h-2}} \left| P_M^{\pi,h-2} - P_{M^\star}^{\pi,h-2} \right| + |\mathcal{A}| \cdot \frac{\alpha}{4|\mathcal{A}| \cdot H}$$

$$= \sum_{s_{h-2}} \left| P_M^{\pi,h-2} - P_{M^\star}^{\pi,h-2} \right| + \frac{\alpha}{4H}$$

By induction on $h$, we have

$$\frac{1}{|\mathcal{A}|} \sum_{s_{h-1}} \sum_{a_{h-1}} \sum_{s_h} P_M^h |P_M^{\pi,h-1} - P_{M^\star}^{\pi,h-1}| = \sum_{s_{h-1}} \left| P_M^{\pi,h-1} - P_{M^\star}^{\pi,h-1} \right| \leq h \cdot \frac{\alpha}{4H} \leq \frac{\alpha}{4}$$

An analogous argument shows that

$$\frac{1}{|\mathcal{A}|} \sum_{s_{h-1}} \sum_{a_{h-1}} \sum_{s_h} |P_{M'}^h(P_{M'}^{\pi,h-1} - P_{M^\star}^{\pi,h-1})| \leq \frac{\alpha}{4}$$

Putting these together we have:

$$\alpha \leq \mathcal{D}(\pi, M, M', h) \leq \frac{1}{|\mathcal{A}|} \sum_{s_{h-1}} \sum_{a_{h-1}} \sum_{s_h} P_M^h |P_M^{\pi,h-1} - P_{M^\star}^{\pi,h-1}|$$

$$+ \frac{1}{|\mathcal{A}|} \sum_{s_{h-1}} \sum_{a_{h-1}} \sum_{s_h} P_{M'}^h |P_{M^\star}^{\pi,h-1} - P_{M'}^{\pi,h-1}| \mathcal{W}(\pi, M, h) + \mathcal{W}(\pi, M', h)$$

$$\leq \frac{\alpha}{4} + \frac{\alpha}{4} + \mathcal{W}(\pi, M, h) + \mathcal{W}(\pi, M', h)$$

$$= \frac{\alpha}{2} + \mathcal{W}(\pi, M, h) + \mathcal{W}(\pi, M', h)$$

Therefore $\mathcal{W}(\pi, M, h) + \mathcal{W}(\pi, M', h) \geq \alpha/2$ and since $\mathcal{W}(\pi, M, h), \mathcal{W}(\pi, M', h) \geq 0$ we have either $\mathcal{W}(\pi, M, h) \geq \frac{\alpha}{4} \geq \frac{\alpha}{4|\mathcal{A}|H}$ or $\mathcal{W}(\pi, M', h) \geq \frac{\alpha}{4} \geq \frac{\alpha}{4|\mathcal{A}|H}$, as desired. $\qquad\square$

**Lemma 2.** *(Explore or Exploit) Suppose the true model $M^\star$ is never eliminated. At iteration $t$, one of the following two conditions must hold: either there exists $M \in \mathcal{M}_t, h_t \leq H$ such that $\mathcal{W}(\pi_{\text{explore}}^t, M, h_t) > \frac{\epsilon}{4H^2|\mathcal{A}|^2}$, or the algorithm returns $\pi_{\text{exploit}}$ such that $v_{\pi_{\text{exploit}}} > v_{\pi^\star} - \epsilon$.*

*Proof.* First consider the case where $v_{\text{explore}}(\pi_{\text{explore}}^t, \mathcal{M}_t) > \frac{\epsilon}{|\mathcal{A}|}$. Then by definition of $v_{\text{explore}}$ there exists some $M, M'$ and $h \in [H]$ such that $\mathcal{D}(\pi_{\text{explore}}^t, M, M', h) > \frac{\epsilon}{H|\mathcal{A}|}$. By Lemma 1 we also have $\mathcal{W}(\pi_{\text{explore}}^t, M, h_t) > \frac{\epsilon}{4H^2|\mathcal{A}|^2}$ or $\mathcal{W}(\pi_{\text{explore}}^t, M', h_t) > \frac{\epsilon}{4H^2|\mathcal{A}|^2}$ for some $h_t \leq h$.

Now consider the case where $v_{\text{explore}}(\pi_{\text{explore}}^t, \mathcal{M}_t) \leq \frac{\epsilon}{|\mathcal{A}|}$. Since $\pi_{\text{exploit}}$ is the optimal policy for $\tilde{M}$, we have $v_{\tilde{M}}^{\pi_{\text{exploit}}} \geq v_{\tilde{M}}^{\pi^\star}$.

We will now bound $|v_{\tilde{M}}^{\pi^\star} - v_{M^\star}^{\pi^\star}|$ :

$$|v_{\tilde{M}}^{\pi^\star} - v_{M^\star}^{\pi^\star}| = \Big| \sum_{h=1}^{H} \sum_{s_h} P_{\tilde{M}}^{\pi^\star, h}(s_h) R^\star(s_h) - \sum_{h=1}^{H} \sum_{s_h} P_{M^\star}^{\pi^\star, h}(s_h) R^\star(s_h) \Big|$$

$$= \Big| \sum_{h=1}^{H} \sum_{s_h} (P_{\tilde{M}}^{\pi^\star, h}(s_h) - P_{M^\star}^{\pi^\star, h}(s_h)) R^\star(s_h) \Big|$$

$$\leq \sum_{h=1}^{H} \sum_{s_h} |P_{\tilde{M}}^{\pi^\star, h}(s_h) - P_{M^\star}^{\pi^\star, h}(s_h)|$$

where we have used the fact that the per-timestep rewards are bounded by 1. Expanding further we get:

$$|v_{\tilde{M}}^{\pi^\star} - v_{M^\star}^{\pi^\star}| \leq \sum_{h=1}^{H} \sum_{s_h} |P_{\tilde{M}}^{\pi^\star, h} - P_{M^\star}^{\pi^\star, h}(s_h)|$$

$$= \sum_{h=1}^{H} \sum_{s_h} \Big| \sum_{s_{h-1}} \sum_{a_{h-1}} P_{\tilde{M}}^{\pi^\star, h-1} \pi^\star(a_{h-1}|s_{h-1}) P_{\tilde{M}}^{h} - P_{M^\star}^{\pi^\star, h-1} \pi^\star(a_{h-1}|s_{h-1}) P_{M^\star}^{h} \Big|$$

$$\leq \sum_{h=1}^{H} \sum_{s_h} \sum_{s_{h-1}} \sum_{a_{h-1}} \Big| P_{\tilde{M}}^{\pi^\star, h-1} \pi^\star(a_{h-1}|s_{h-1}) P_{\tilde{M}}^{h} - P_{M^\star}^{\pi^\star, h-1} \pi^\star(a_{h-1}|s_{h-1}) P_{M^\star}^{h} \Big|$$

$$\leq \sum_{h=1}^{H} \sum_{s_h} \sum_{s_{h-1}} \Big| P_{\tilde{M}}^{\pi^\star, h-1} P_{\tilde{M}}^{h} - P_{M^\star}^{\pi^\star, h-1} P_{M^\star}^{h} \Big| \cdot |\mathcal{A}|$$

$$\leq \sum_{h=1}^{H} \mathcal{D}(\pi^\star, \tilde{M}, M^\star, h)|\mathcal{A}|$$

Note that $\sum_{h=1}^{H} \mathcal{D}(\pi^\star, \tilde{M}, M^\star, h) \leq v_{\text{explore}}(\pi^\star, \mathcal{M}_t) \leq v_{\text{explore}}(\pi_{\text{explore}}^t, \mathcal{M}_t) \leq \frac{\epsilon}{|\mathcal{A}|}$ since $\pi_{\text{explore}}^t$ is the optimal policy for the exploration MDP. Therefore we have

$$|v_{\tilde{M}}^{\pi^\star} - v_{M^\star}^{\pi^\star}| \leq \frac{\epsilon}{|\mathcal{A}|} \cdot |\mathcal{A}| = \epsilon$$

Combining this with the fact that $v_{\widetilde{M}}^{\pi_{\text{exploit}}} \geq v_{\widetilde{M}}^{\pi^\star}$, we get $v_{M^\star}^{\pi_{\text{exploit}}} \geq v_{M^\star}^{\pi_{\text{exploit}}} - \epsilon$.

$\square$

The proof for the following lemma can be found in [49] (Lemma 8).

**Technical Lemma 3.** *Suppose that $|\widetilde{\mathcal{W}}(\pi_{\text{explore}}^t, M, h_t) - \mathcal{W}(\pi_{\text{explore}}^t, M, h_t)| \leq \phi$ holds for all $t, h_t$ and $M \in \mathcal{M}$. Then:*

1. *$M^\star \in \mathcal{M}_t$ for all $t$.*

2. *Denote $\widetilde{\mathcal{M}}_{t+1} = \{M \in \widetilde{\mathcal{M}}_t : A_{h_t}(\pi_{\text{explore}}^t, M) \leq 2\phi\}$ with $\widetilde{\mathcal{M}}_1 = \mathcal{M}$. We have $\mathcal{M}_t \subseteq \widetilde{\mathcal{M}}_t$ for all $t$.*

**Lemma 3.** *(Iteration Complexity) Let $d = \max_{1 \leq h \leq H} rank(A_h)$ and $\phi = \frac{\epsilon}{24H^2|\mathcal{A}|^2\sqrt{d}}$. Suppose that $|\widetilde{\mathcal{W}}(\pi_{\text{explore}}^t, M, h) - \mathcal{W}(\pi_{\text{explore}}^t, M, h)| \leq \phi$ holds for all $t$, $h \leq H$ and $M \in \mathcal{M}$. Then the number of rounds of Algorithm 1 with the `UpdateModelSet` routine given by Algorithm 2 is at most $Hd \log(\frac{\beta}{2\phi})/\log(5/3)$.*

*Proof.* From Lemma 2, if the algorithm does not terminate then we have $\pi_{\text{explore}}^t, h_t, M' \in \mathcal{M}_t$ such that:

$$\mathcal{W}(\pi_{\text{explore}}^t, M', h_t) > \frac{\epsilon}{4H^2|\mathcal{A}|^2} = 6\sqrt{d}\phi$$

which can be rewritten as:

$$A_{h_t}(\pi_{\text{explore}}^t, M') = U_{h_t}(\pi_{\text{explore}}^t)^\top V_{h_t}(M') > 6\sqrt{d}\phi.$$

For any $h$ and $t$, denote $O_t^h$ as the origin-centered minimum volume enclosing ellipsoid (MVEE) of $\{V_h(M) : M \in \widetilde{\mathcal{M}}_t\}$. Also denote $O_{t,+}^{h_t}$ as the origin-centered MVEE of $\{v \in O_t^{h_t} : U_{h_t}(\pi_{\text{explore}}^t))^\top v \leq 2\phi\}$. Note that by definition of $\widetilde{\mathcal{M}}_{t+1}$, for all $M \in \widetilde{\mathcal{M}}_{t+1}$ we have $A_{h_t}(\pi_{\text{explore}}^t, M) = U_{h_t}(\pi_{\text{explore}}^t)^\top V_{h_t}(M) \leq 2\phi$ and since $O_{t+1}^{h_t} \subseteq O_t^{h_t}$ we have $O_{t+1}^{h_t} \subseteq O_{t,+}^{h_t}$ and hence $\text{vol}(O_t^{h_t}) \leq \text{vol}(O_{t,+}^{h_t})$. See Figure 3 for an illustration.

We can then apply Lemma 11 in [21], (setting $B := O_t^h, p := U_{h_t}(\pi_{\text{explore}}^t), v := V_{h_t}(M'), \kappa := 6\sqrt{d}\phi$), and get:

$$\frac{\text{vol}(O_{t+1}^{h_t})}{\text{vol}(O_t^{h_t})} \leq \frac{\text{vol}(O_{t,+}^{h_t})}{\text{vol}(O_t^{h_t})} \leq 3/5$$

This shows that if the algorithm does not terminate, then we shrink the volume of $O_t^{h_t}$ by a constant factor. To show that the number of iterations is small, we must now show that the initial volume is not too large and the final volume is not too small. Denote $\Phi := \sup_{\pi \in \Pi} \|U_{h_t}(\pi)\|_2$ and $\Psi := \sup_{M \in \mathcal{M}} \|V_{h_t}(M)\|_2$. For $O_1^h$, we have that $\text{vol}(O_1^h) \leq c_d \Psi^d$ where $c_d$ is the volume of the unit Euclidean ball in $d$ dimensions. For any $t$, we have

$$O_t^h \supseteq \{q \in \mathbb{R}^d : \max_{p:\|p\|_2 \leq \Phi} q^\top p \leq 2\phi\} = \{q \in \mathbb{R}^d : \|q\|_2 \leq 2\phi/\Phi\}$$

Hence, at termination we must have that $\text{vol}(O_T^h) \geq c_d(2\phi/\Phi)^d$. Using the volume of $O_T^h$ and the lower bound of the volume of $O_T^h$ and the fact that every round we shrink the volume of $O_t^{h_t}$ by a constant factor, we must have that for any $h \in [H]$ the number of rounds for which $h_t = h$ is

Figure 3: Illustration of geometric argument for $d = 2$. Black dots represent embeddings of the models in $\mathcal{M}$, the star represents the embedding of the exploration policy $\pi_{\text{explore}}^t$.

at most $d \log(\frac{\Phi\Psi}{2\phi})/\log(5/3)$. Using the definition $\beta \geq \Phi\Psi$, this gives an iteration complexity of $Hd \log(\frac{\beta}{2\phi})/\log(5/3)$.

$\square$

**Theorem 1.** *Assuming that $M^\star \in \mathcal{M}$, for any $\epsilon, \delta \in (0,1]$ set $\phi = \frac{\epsilon}{24H^2|\mathcal{A}|^2\sqrt{d}}$ and denote $T = Hd \log(\frac{\beta}{2\phi})/\log(5/3)$. Run Algorithm 1 with inputs $(\mathcal{M}, n, \phi)$ where $n = \Theta(H^4|\mathcal{A}|^4 d \log(T|\mathcal{M}|/\delta)/\epsilon^2)$, and the `UpdateModelSet` routine given by Algorithm 2. Then with probability at least $1 - \delta$, Algorithm 1 outputs a policy $\pi_{\text{exploit}}$ such that $v_{\pi_{\text{exploit}}} \geq v^* - \epsilon$. The number of trajectories collected is at most $\tilde{O}\left(\frac{H^5 d^2 |\mathcal{A}|^4}{\epsilon^2} \log\left(\frac{T|\mathcal{M}||\Pi|}{\delta}\right)\right)$.*

*Proof.* We condition on the event that $|\widetilde{\mathcal{W}}(\pi_{\text{explore}}^t, M, h) - \mathcal{W}(\pi_{\text{explore}}^t, M, h)| \leq \phi$ for all $t$ and $h \in [H], M \in \mathcal{M}$. Under this condition, by Lemma 3 we know that the algorithm must terminate in at most $Hd \log(\frac{\beta}{2\phi})/\log(5/3)$ iterations. Once the algorithm terminates, we know that we must have an $\epsilon$-optimal policy by Lemma 2. Now we show that this condition holds with probability at least $1 - \delta$. Applying Technical Lemma 2 and performing a union bound over all $h \in \{1, ..., H\}$ and $t \in \{1, ..., T\}$, we have that with probability at least $1 - \delta$:

$$\left|\widetilde{\mathcal{W}}(\pi_{\text{explore}}^t, M, h) - \mathcal{W}(\pi_{\text{explore}}^t, M, h)\right| \leq \frac{4 \log(4TH|\mathcal{M}||\Pi|/\delta)}{3n} + 4\sqrt{\frac{\log(4TH|\mathcal{M}||\Pi|/\delta)}{n}}$$
$$\leq 8\sqrt{\frac{\log(4TH|\mathcal{M}||\Pi|/\delta)}{n}}$$

for all $T$ iterations of the algorithm and $n > 4 \log(4TH|\mathcal{M}||\Pi/\delta)/3$. Requiring this upper bound to be less than $\phi = \frac{\epsilon}{24H^2|\mathcal{A}|^2\sqrt{d}}$ and solving for $n$, we get:

$$8\sqrt{\frac{\log(4TH|\mathcal{M}||\Pi|/\delta)}{n}} \leq \frac{\epsilon}{24H^2|\mathcal{A}|^2\sqrt{d}}$$

$$64\frac{\log(4TH|\mathcal{M}||\Pi|/\delta)}{n} \leq \frac{\epsilon^2}{576H^4|\mathcal{A}|^4 d}$$

$$\frac{36864H^4|\mathcal{A}|^4 d\log(4TH|\mathcal{M}||\Pi|/\delta)}{\epsilon^2} \leq n$$

Since we are sampling this number of trajectories at each iteration of the algorithm, the total number of trajectories is therefore $n \cdot T = \tilde{\mathcal{O}}(\frac{H^5 d^2|\mathcal{A}|^4}{\epsilon^2}\log(\frac{T|\mathcal{M}||\Pi|}{\delta}))$. $\qquad\square$

### A.3 Extension to Unknown $d$

---

**Algorithm 3** $(\mathcal{M}, \Pi, \epsilon, \delta)$

---

1: **for** $i = 1, 2, ...$ **do**
2:     Set $d_i \leftarrow 2^i$
3:     Set $\delta_i \leftarrow \frac{\delta}{i(i+1)}$
4:     Set $\phi_i \leftarrow \frac{\epsilon}{24H^2|\mathcal{A}|^2\sqrt{d_i}}$
5:     Set $n_i = \frac{36864H^4|\mathcal{A}|^4 d_i \log(4TH|\mathcal{M}||\Pi|/\delta_i)}{\epsilon^2}$
6:     Run DREEM$(\mathcal{M}, \Pi, n_i, \epsilon, \phi_i)$ until it returns a policy $\pi$ or $t > Hd_i\log(\frac{\beta}{2\phi_i})/\log(5/3)$
7:     **if** a policy $\pi$ was returned **then**
8:         Return $\pi$
9:     **end if**
10: **end for**

---

Algorithm 3 shows how a near-optimal policy can be computed without requiring knowledge of the $d$ parameter. It operates by running DREEM as a subroutine using guesses for $d$ which follow a doubling schedule with adjusted values of the $\delta$ parameter.

First note that since we assign $\frac{\delta}{i(i+1)}$ failure probability to each round of Algorithm 3, the total probability that any of the subroutines returns a suboptimal policy is $\sum_{i=1}^{\infty}\frac{\delta}{i(i+1)} = \delta\sum_{i=1}^{\infty}(\frac{1}{i} - \frac{1}{i+1}) = \delta$. Also note that $M^\star$ is never eliminated. Therefore with probability $1 - \delta$, if the algorithm does return a policy, it is near optimal. It remains to show that Algorithm 3 terminates. We know that the subroutine terminates with a near-optimal policy when we reach the first iteration $i$ such that $d \leq d_i = 2^i$. Then we must have $d_{i-1} < d \leq d_i \implies 2d_{i-1} < 2d \leq 2d_i \implies d_i \leq 2d \implies 2^i \leq 2d \implies i \leq \log_2 d + 1$, so the algorithm terminates after $\log_2 d + 1$ iterations. The sample complexity of each subroutine call is monotonically increasing, and the sample complexity of the last call is $\tilde{\mathcal{O}}(\frac{H^5 d_i|\mathcal{A}|^4}{\epsilon^2}\log(\frac{T|\mathcal{M}|}{\delta_i})) = \tilde{\mathcal{O}}(\frac{H^5 2d|\mathcal{A}|^4}{\epsilon^2}\log(\frac{(\log d)^2 T|\mathcal{M}|}{\delta})) = \tilde{\mathcal{O}}(\frac{H^5 d|\mathcal{A}|^4}{\epsilon^2}\log(\frac{T|\mathcal{M}|}{\delta}))$, where we have suppressed constant factors and factors which are logarithmic in $d$ at the last step. Combining this with the fact that there are at most $\log_2 d + 1$ iterations, we see that the sample complexity of Algorithm 3 is the same as Algorithm 1 up to factors which are logarithmic in $d$.

## B Practical Algorithm Details

### B.1 Model Updates

For the environments with deterministic dyanamics (Maze and Continuous Control), we found it helpful to train the models to make multi-step rather than single-step predictions. For a trajectory $\tau = (s_i, a_i, s_{i+1}, a_{i+1}, ...s_{i+K+1}) \in \mathcal{R}$ and model $M$ with parameters $\theta$, the loss is given by:

$$\mathcal{L}(\theta, \tau) = \sum_{j=1}^{K} \|s_{i+j+1} - M_\theta(\tilde{s}_{i+j}, a_{i+j})\|_2^2 \text{ such that } \tilde{s}_{i+j} = \begin{cases} s_i & \text{if } j = 0 \\ M_\theta(\tilde{s}_{i+j-1}, a_{i+j-1}) & \text{else} \end{cases}$$

Beyond the first step, the model takes as input its prediction from the previous step, and gradients are backpropagated through the model unrolled over $K$ time steps. This helps the models make more robust predictions over longer timescales, since errors which are magnified over time get penalized and the models are trained on noisy inputs. We also used a simple form of prioritized experience replay [41], where we sample trajectories from the last epoch with higher probability ($p = 0.5$) and from all remaining epochs uniformly. This helps the models quickly learn from recent experience. For the stochastic environment (combination lock), we found that single-step predictions worked well.

### B.2 Planning

#### B.2.1 Deterministic Dynamics

Algorithm 4 shows the procedure for searching in a continuous state space when the dynamics are deterministic (note the start state can still be stochastic). If the state space is discrete, exponential time complexity can be avoided by marking states as visited and only expanding unvisited states, an idea which is used in breadth-first or depth-first search. Here we generalize this idea for continuous spaces using a priority queue, where expanded states are assigned a priority based on their minimum distance to other states in the currently expanded search tree. If two action sequences lead to nearby states, only one of these states is likely to be expanded given a fixed computational budget as the other will be given low priority due to its proximity to the first. The algorithm returns variable length action sequences, and may be called multiple times within an episode.

---

**Algorithm 4** $\texttt{DeterministicPlanner}(s, \mathcal{M}, N_{\max}, \texttt{mode})$

---
1: **Input** Set $\mathcal{M} = \{f_{\theta_1}, ..., f_{\theta_E}\}$ of dynamics models, current state $s$, max graph size $N_{\max}$.
2: Define root node: for $i = 1, ..., E$ set $v.s_i = s$
3: Set $v.\hat{s} \leftarrow s$
4: Set $v.priority \leftarrow \infty, v.\pi \leftarrow []$
5: Initialize graph $\mathcal{V} \leftarrow \{v\}$
6: **while** $|\mathcal{V}| < N_{\max}$ **do**
7:      Pick vertex to expand: $v \leftarrow \text{argmax}_{v \in \mathcal{V}} \left[ v.priority \right]$
8:      Set $v.priority \leftarrow -\infty$
9:      **for** $a \in \mathcal{A}$ **do**
10:        **if** $\texttt{mode} = \texttt{explore}$ **then**
11:          Utility is maximum disagreement: $u \leftarrow \max_{f_{\theta_i}, f_{\theta_j} \in \mathcal{M}} \|f_{\theta_i}(v.s_i, a) - f_{\theta_j}(v.s_j, a)\|_2^2$
12:        **else if** $\texttt{mode} = \texttt{exploit}$ **then**
13:          Utility is average predicted reward: $u \leftarrow \frac{1}{E} \sum_{i=1}^{E} R^\star(f_{\theta_i}(v.s_i, a))$
14:        **end if**
15:        Define new node $v'$ with $v'.\pi \leftarrow append(v.\pi, a)$
16:        For $i = 1, ..., E$, set $v'.s_i \leftarrow f_{\theta_i}(v.s_i, a)$
17:        Set $v'.\hat{s} \leftarrow \frac{1}{E} \sum_{i=1}^{E} v'.s_i$
18:        Set $v'.priority \leftarrow \min_{v \in \mathcal{V}} \|v'.\hat{s} - v.\hat{s}\|_2$
19:        Set $v'.utility \leftarrow v.utility + u$
20:        $\mathcal{V} \leftarrow \mathcal{V} \cup \{v'\}$
21:      **end for**
22: **end while**
23: $v^\star \leftarrow \text{argmax}_{v \in \mathcal{V}} v.utility / |v.\pi|$
24: Return $v^\star.\pi$

---

#### B.2.2 Stochastic Dynamics

When planning in a stochastic environment, we use Monte-Carlo Tree Search where a given node $\nu$ in the tree at depth $h$ corresponding to a fixed action sequence $\pi_A$ (of length $h$) consists of empirical

estimates of $P_{M_1}^{\pi_A,h}, ..., P_{M_E}^{\pi_A,h}$ for each model in the ensemble. Concretely, $\nu$ is represented as a tensor of size $|E| \times K \times m$ where $|E|$ is the number of models in the ensemble, $K$ is the number of samples drawn from each model $M$ to estimate its predicted distribution $P_M^{\pi_A,h}$, and $m$ is the dimension of the state vector. The root node is initialized with the current state $s$, i.e. $S_{e,k}^{\text{root}} = s$ for all $1 \le e \le E, 1 \le k \le K$. Given an action $a \in \mathcal{A}$ applied at a node $\nu$, the next node is computed as follows: $\nu'_{e,k} \sim M_e(\nu_{e,k}, a)$.

The rewards at each node, which are then used to choose which action to execute in the real environment, depend on whether the algorithm is in explore or exploit mode. In explore mode, the reward is given by $R_{\text{explore}}(\nu) = \max_{1 \le e,e' \le E} \|\hat{P}_{M_e}(\cdot) - \hat{P}_{M_{e'}}(\cdot)\|_{TV}$, where $\hat{P}_{M_e}(\cdot)$ is the empirical distribution computed using the $K$ samples $\nu_{e,:}$ drawn from model $M_e$. In exploit mode, the reward is given by $R_{\text{exploit}}(\nu) = \frac{1}{E \cdot K} \sum_{e=1}^{E} \sum_{k=1}^{K} R^\star(\nu_{e,k})$, i.e. the mean reward across all samples and all models in the ensemble. After a fixed number of playouts, the MCTS procedure returns a sequence of actions which maximizes the expected exploration or exploitation reward. We execute the first action in this sequence, and then replan at every step. See our code release for full details.

# C   Experiment Details

For Neural-E$^3$, we found that specifying a number of exploration epochs was simpler than tuning the $\epsilon$ parameter in Algorithm 1, which determines when to switch to the exploit phase and which is task-dependent. This is listed in the table of hyperparameters.

## C.1   Stochastic Combination Lock

The environment consists of $H$ levels with 3 underlying states per level (denoted $s_{1,h}, s_{2,h}, s_{3,h}$) and 4 possible actions. The states $s_{3,:}$ are dead states from which it is impossible to recover: all actions from $s_{3,h}$ lead to $s_{3,h+1}$ with probability 1. 2 actions lead from each of the states $s_{1,h}$ and $s_{2,h}$ to the dead state $s_{3,h+1}$, and the other two actions lead to one of $s_{1,h+1}$ and $s_{2,h+1}$. Which action leads to which state is randomly determined when the environment is initialized and kept fixed thereafter. This means that simply repeating a single action is unlikely to lead to the reward. With probability $\alpha = 0.1$, the effect of the actions leading to the good states at the next level is flipped. Therefore, executing a preplanned action sequence without accounting for intermediate observations is likely to lead to the dead states.

**Standard Reward Variant:** The reward is zero everywhere except at the last two states, where a reward of 5 is given for one of the actions.

**Antishaped Reward Variant:** As above, a reward of 5 is given at the last two states for one of the actions. Furthermore, a reward of 0.1 is given for transitioning to any of the dead states (for example, from $s_{1,h}$ to $s_{3,h+1}$), and a negative reward of $-1/H$ is given for transitioning to any state which is not a dead state (for example, from $s_{1,h}$ to $s_{1,h+1}$). This means that until the agent has explored the last states which give high reward, the locally optimal policy appears to be to transition to the dead states as quickly as possible.

Table 1: DQN Hyperparameters

| Hyperparameter | Values Considered | Final Value |
|---|---|---|
| Learning Rate | $0.01, 0.001, 0.0001$ | $0.01$ |
| Hidden Layer Size | $64$ | $64$ |
| Prioritized Replay | `true` | `true` |
| Discount Factor | $0.99$ | $0.99$ |
| Exploration Fraction (episodes) | $\{0.1, 0.01, 0.001\}$ | $0.001$ for standard rewards $0.01$ for antishaped rewards |

Figure 4 shows results for both variants of the task for larger numbers of episodes. A somewhat surprising result was that for the standard variant of the task, the DQN is still able to achieve good performance for longer horizons, using much fewer samples than PPO+RND. We found that the DQN performed best when the exploration fraction is set to be very low (0.001 as shown in Table 1),

(a) With standard rewards

(b) With antishaped rewards

Figure 4: Results on the Stochastic Combination Lock task given more episodes. PPO+RND is able to eventually achieve reasonable performance given enough episodes.

meaning that the DQN agent quickly begins to act greedily. This suggests that acting greedily leads the agent to explore the environment better than uniform exploration. Uniform exploration leads to a vanishingly small chance of reaching the reward ($\approx 10^{-6}$ for $H = 20$). One explanation could be that the network happens to be initialized in a manner that gives optimistic estimates for the Q-values. We found that the DQN performance was highly dependent on implementation details, for example, the implementation in [43] gave very poor results, as did removing the prioritized experience replay.

Table 2: PPO+RND Hyperparameters

| Hyperparameter | Values Considered | Final Value |
|---|---|---|
| Learning Rate | $0.01, 0.001, 0.0001$ | $0.001$ |
| Hidden Layer Size | $64$ | $64$ |
| $\gamma_I$ | $0.99$ | $0.99$ |
| $\gamma_E$ | $0.999$ | $0.999$ |
| $\lambda$ | $0.95$ | $0.95$ |
| Intrinsic Reward coefficient | $1.0$ | $1.0$ |
| Extrinsic Reward coefficient | $2, 100$ | $100$ |

Table 3: $E^3$ Hyperparameters

| Hyperparameter | Values Considered | Final Value |
|---|---|---|
| Learning Rate | $0.01, 0.001$ | $0.01$ |
| Hidden Layer Size | $50, 100$ | $50$ |
| Ensemble Size | $5, 10$ | $5$ |
| Minibatch Size | $100$ | $100$ |
| Number of Exploration Epochs | $25H, 50H, 75H$ | Horizon-dependent: $H = 5 : 25H, H = 10 : 50H$ $H = 15 : 50H, H = 20 : 75H$ |
| Exploration Episodes per Epoch | $1$ | $1$ |
| Model Updates per Epoch | $100$ | $100$ |
| MCTS playouts | $200$ | $200$ |
| MCTS samples ($K$) | $100$ | $100$ |

For Neural-$E^3$, we found that training a DQN offline using the data collected in the replay buffer (as described in Section 4.3) performed better than using MCTS to maximize the reward, especially on the task variant with antishaped rewards. This is likely because MCTS biases the search tree towards action sequences which accumulate the best reward so far, and so the misleading rewards can lead the

search procedure away from action sequences which produce the globally optimal reward. All the Neural-E$^3$ results reported use the DQN exploitation method.

## C.2 Maze Domain

We used the source code for the maze environment provided by the authors `https://github.com/junhyukoh/value-prediction-network`, and set the number of goals to 1 and the time limit to 100. All results are reported using 3 random seeds.

The forward dynamics model architecture is a 3-layer convolutional network (1 convolutional layer followed by 2 deconvolutional layers, all with 16 feature maps). Actions are embedded to a 16-dimensional vector replicated across all spatial locations and added to the feature maps. A separate reward head consists of 2 strided convolution layers followed by a fully-connected layer producing a scalar.

Table 4: DQN Hyperparameters

| Hyperparameter | Values Considered | Final Value |
|---|---|---|
| Learning Rate | $10^{-3}, 10^{-4}, 10^{-5}$ | $10^{-4}$ |
| Feature Maps | $8, 32$ | 8 |
| Convolutional Layers | $1, 2, 3$ | 1 |
| Hidden Layer Size | $64, 256$ | 64 |
| Prioritized Replay | `true` | `true` |
| Parameter Noise | `false` | `false` |
| Discount Factor | 0.99 | 0.99 |

Table 5: $E^3$ Hyperparameters

| Hyperparameter | Values Considered | Final Value |
|---|---|---|
| Learning Rate | $10^{-3}, 10^{-4}$ | $10^{-3}$ |
| Number of Feature Maps | 16 | 16 |
| Hidden Layer Size | 64 | 64 |
| Ensemble Size | $4, 8$ | 4 |
| Minibatch Size | 64 | 64 |
| Number of exploration Epochs | $5, 10$ | 5 |
| Exploration Episodes per Epoch | 10 | 10 |
| Model Updates per Epoch | 10000 | 10000 |
| Unrolling steps ($K$) | $10, 20$ | 10 |
| Maximum Graph Size during Planning ($N_{\max}$) | $500, 1000, 2000, 5000$ | 2000 |

## C.3 Continuous Control Domains

We used the environments provided by OpenAI Gym [6], available at: `https://gym.openai.com/envs/#classic_control`. In initial experiments we experimented with adding parameter noise to the DQN, but found that this did not help.

Table 6: DQN Hyperparameters

| Hyperparameter | Values Considered | Final Value |
|---|---|---|
| Learning Rate | $10^{-2}, 10^{-3}, 10^{-4}$ | $10^{-3}$ |
| Hidden Layer Size | $64, 256$ | 64 |
| Prioritized Replay | `true, false` | `true` |
| Parameter Noise | `true, false` | `false` |
| Discount Factor | 0.99 | 0.99 |

The forward model architecture is a 3-layer MLP with LeakyReLU non-linearities. The action is embedded to a vector of size 64 and multiplied component-wise with the first layer of hidden units. All models are trained using Adam [25].

Figure 5: Predictions by the different dynamics models in the ensemble for the Maze task, 29 steps into the future (best viewed in color). The green dot is the agent and the blue dot is the goal. The models all agree in their predictions up to steps 27 and 28, but disagree for step 29 where the agent collects the reward.

Table 7: PPO+RND Hyperparameters

| Hyperparameter | Values Considered | Final Value |
|---|---|---|
| Learning Rate | $10^{-3}, 10^{-4}, 10^{-5}$ | $10^{-4}$ |
| Hidden Layer Size | 64 | 64 |
| $\gamma_I$ | 0.99 | 0.99 |
| $\gamma_E$ | 0.999 | 0.999 |
| $\lambda$ | 0.95 | 0.95 |
| Intrinsic Reward coefficient | 1.0 | 1.0 |
| Extrinsic Reward coefficient | 2 | 2 |

Table 8: $E^3$ Hyperparameters

| Hyperparameter | Values Considered | Final Value |
|---|---|---|
| Learning Rate | $10^{-3}, 10^{-4}$ | $10^{-4}$ |
| Hidden Layer Size | 64 | 64 |
| Ensemble Size | 8 | 8 |
| Minibatch Size | 64 | 64 |
| Number of exploration Epochs | 10 | 10 |
| Exploration Episodes per Epoch | $\{10, 20\}$ | 10 |
| Model Updates per Epoch | 2000 | 2000 |
| Unrolling steps ($K$) | 20 | 20 |
| Maximum Graph Size during Planning ($N_{\max}$) | 2000 | 2000 |
| DQN Learning Rate | $10^{-2}, 3 \cdot 10^{-3}, 1 \cdot 10^{-3}, 3 \cdot 10^{-4}, 1 \cdot 10^{-4}$ | $3 \cdot 10^{-4}$ |
| DQN Updates for Exploit Phase | $\{500000, 750000, 1000000\}$ | 750000 |
| DQN Target Network Update Frequency | 5000 | 5000 |

For the exploit phase, we initially train a DQN for 750000 updates on the data collected from the replay buffer. It is then continued to be trained, and if performance begins decreasing, the model is reverted to its best performing set of weights and training is stopped.