[Reviews · NeurIPS 2019]

Reviewer 1



After rebuttal: Thanks the authors for addressing my concerns. I have read the authors feedback and other reviews. I'll keep my original score. ------------------------ This work proposes an E3-style model-based algorithm for the finite horizon MDPs with known reward. The agent is able to collect data when running the algorithm and the goal is to find a near optimal policy. Early work includes [4], [21], and [39]. Using ranks of error matrices to represent complexity and some proof techniques are related to [18] and [41]. The paper is technically sound and clearly written. In the theoretical side, the authors prove a polynomial sample complexity bound in terms of |A|, H, and the rank of the model misfit matrix, thus avoiding the dependence on |S|. This work is also supported by the experiments. The empirical result shows that Neural E3 algorithm can achieve similar progress as standard algorithms. Some comments: 1. In general, I like the idea of this work. The algorithm utilizes the uncertainty over predicted states instead of directly maximizing the expected return. In the exploration step, the algorithm finds a policy that leads to mismatch between two models. Therefore at least one model is not the true model. The algorithm will then collect enough trajectories of data to eliminate the inconsistent model. 2. This paper assumes that the true reward R* is known and it seems to be too strong. Predicting future states include predicting future rewards when the true reward is known. My major concern is that the idea of considering the mismatch between probability distribution between models only works under this assumption. I was wondering whether this work can be extended to the case when R* is unknown. 3. The sample complexity result is important. However, the analysis might not be novel since some are closely related to and adapted from [41]. 4. The experiment further strengthens this work. Closely related work [18] and [41] do not provide practical instantiations. In some simple tasks, the proposed Neural E3 algorithm achieves similar results as baselines. However, the Neural E3 works only in the deterministic environment so there is still a gap between the theory and the experiment. Is it possible to extend the Neural E3 to stochastic environment?

Reviewer 2



UPDATE after rebuttal ====== Thanks for the author response. I read the response and other reviews as well. In response, the authors explained the discrepancy between the idealized version and the practical version of algorithm. I marginally tend to accept this work. Quality ====== The theoretical analysis is sound for the problem and the proposed algorithm. Under the realizability and optimal planning assumptions, the authors prove that algorithm 1 and algorithm 2 can find near-optimal policy in polynomial sample complexity with factored MDP. The experiments show that the proposed practical algorithm (though may not exactly match the ideal algorithm analyzed) can overperform previous method. Clarity ====== The organization of the lemmas and theorems is good, and the statement is clear, precise and relevant. The model elimination part for the practical method is seemly not clear according to the paper. The authors tell us the methods to update model accuracy in practice. The model elimination stage seems missing in the practical algorithm (model update as in Sec.4.1 instead of model elimination?), which makes it unclear how the theoretical analysis could guide practical exercise. Originality ====== The paper extends the classical Explicit Explore-Exploit(E^3) Algorithm to large or infinite state spaces. The use of maximum disagreement as uncertainty measure is novel for exploration. Although there is some literature (‘MAX’ and [Deepak2019]) also using disagreement among ensemble models for exploration, the methods for disagreement measurement in this paper is different. However, for large state space, the author uses the structural properties in factored MDP, which is well-studied. Also, it seems that the technical part of proof follows similarly as in [41]. [Deepak2019] Deepak Pathak, Dhiraj Gandhi, Abhinav Gupta. Self-Supervised Exploration via Disagreement. In ICML 2019. Significance ====== The theoretical result is well enough to point out that the maximum disagreement as uncertainty measure is sample efficient for exploration and can led to find near-optimal policies in model-based RL. However, the bound seems not as sharp compared to the sample complexity of other model-based algorithms e.g. UCRL. On the other hand, the empirical results show that the proposed method has superoir performance than previous ones. However, my main concern is that there is still a gap between the theoretical algorithm and the practical implementation (especially in continuous control), which should not be neglected.

Reviewer 3



This paper is well presented, and it also has a theoretical justification of the sample efficiency. Main questions: - How did you set the parameter \phi in Alg. 2? which was not presented in line 8 of Alg. 1? - From theorem 1, the sample efficiency depends on |A|^4. Is it possible to improve the algorithm such that it can also handle large/infinite action spaces?

[Author Response · NeurIPS 2019]



Figure 1: Results on stochastic combination lock environment for different horizon lengths. Solid lines represent median across 5 seeds, shaded regions represent range between best and worst seeds.

**Reviewer 1:** Thank you for the review. The assumption that $R^\star$ is known is for compactness of notation and does not restrict the class of problems the algorithm can handle - we will clarify this in the paper. For example, consider a setting where states are represented by $n$-dimensional vectors. We can construct an augmented state space where vectors have $n+1$ dimensions, where the last dimension represents the reward. In this case $R^\star$ simply reads off the last component of the state vector, $R^\star(s) = s[n+1]$, and accurately predicting the reward in the original formulation amounts to accurately predicting the last component of the state vector in this new formulation. In our experiments, we train the model to predict the reward as well as the next state, and do not assume the reward function is known beforehand.

As suggested in the improvements section, we added a new set of experiments in a stochastic environment, the stochastic combination lock in [1]. This is a hard exploration problem where the chance of getting reward with random exploration is exponentially small in horizon length. We will briefly describe the stochastic version of the practical algorithm here, and in detail in the updated paper. The models in the ensemble output the parameters of a distribution over next states rather than single predictions. For each model $M$ we estimate the distribution $P_M^{\pi,h}(\cdot)$ using $k$ points at each step in the rollout, and compute the disagreement-based signal used to drive the exploration by taking the total variation or KL divergence between the empirical distributions for each pair of models. These $k$ points are then used as inputs to the model at the next step in the rollout. We use Monte-Carlo Tree Search as a planner and only execute the first action in the returned sequence, replanning at every step. The results are shown in Figure 1. We see that as the horizon length increases, the performance of the baseline methods degrades whereas Neural-E$^3$ maintains good performance.

**Reviewer 2:** Thank you for the review. We will clarify the relationship between the idealized version of the algorithm and the practical version both here and in the paper. For both algorithms, $\mathcal{M}_t$ represents a set of models which are consistent with the experience gathered so far, i.e. have low error on the current replay buffer. For the idealized algorithm, models which have high error are eliminated through the explicit elimination step in Algorithm 2. For the practical algorithm, models which have high error are avoided by the optimization procedure, and are thus are unlikely to be present in $\mathcal{M}_t$. The main difference between the two is that the idealized version maintains *all* models in the model class which have low error, whereas the practical version only maintains a subset due to time/memory constraints.

Both versions of the algorithm are based on two key ideas: i) computing exploration policies designed to induce disagreement between plausible models ii) doing so internally, without having to interact with the environment. The motivation is that every time the agent interacts with the environment, there is a high chance that the experience gathered will be useful for refining the set of models. The theoretical analysis makes this intuition precise, and explains why disagreement between two plausible models implies error with respect to the true model and hence useful experience (this applies to the practical algorithm as well). While the practical algorithm does approximate the full version space by an ensemble, our experiments suggest that this approximation can often be sufficient in practice.

To clarify the point "the author uses the structural properties in factored MDP, which is well-studied": note that our sample complexity result depends on the rank of the misfit matrix, which is indeed low for factored MDPs but may also be low in other settings (for example, it is also low for low-rank MDPs).

**Reviewer 3:** Thank you for the review. To answer your first question: to get an $\epsilon$-optimal policy, $\phi$ is set according to the formula in Theorem 1, i.e. $\phi = \frac{\epsilon}{24H^2|\mathcal{A}|^2\sqrt{d}}$. Note that this requires knowledge of the rank of the misfit matrix $d$, which may not be known beforehand. In this case one can use a doubling trick to estimate $d$ while still maintaining polynomial sample complexity, as in [18, 41]. We will add this in the updated paper. Concerning the extension to infinite action spaces, that is a very interesting research question which we hope to investigate in future work. There has been some work on infinite action spaces in the contextual bandit setting, however extending such results to multi-step RL is to our knowledge still an open problem. We have added experiments for an additional (stochastic) environment as suggested in the improvements section, please see Figure 1 and our reponse to Reviewer 1.

**References** [1]: *Provably efficient RL with Rich Observations via Latent State Decoding*, Du et. al, (ICML 2019).

[Meta-Review · NeurIPS 2019]

Reviewers find the ideas of exploration to find high disagreement between the dynamic models interesting and appreciate the sample complexity analysis, albeit the proof follows prior work closely. The empirical evaluation is strong enough to support the proposed algorithm in practical environments.